# Serotonergic receptor binding in the brainstem in the Sudden Infant Death Syndrome in a high-risk population

Hannah C. Kinney[1], Rebecca D. Folkerth[2], Morgan E. Nelson[3], Lucy Brink[4], Felicia L. Trachtenberg[5], Jyoti Angal[6], Kevin G. Broadbelt[1], Theonia K. Boyd[7], Elsie H. Burger[8], Jean A. Coldrey[4], Kevin J. Cummings[9], Jhodie R. Duncan[1,10], Amy J. Elliott[6], William P. Fifer[11], Howard J. Hoffman,[12], James C. Leiter[13], Eugene E. Nattie[13], Laura L. Nelsen[14], Hein J. Odendaal[4], David S. Paterson[1], Bradley B. Randall [15], Drucilla J. Roberts[16], Pawel Schubert[17], Mary Ann Sens[18], Shabbir A. Wadee[19], Colleen Wright[17,20], Dan Zaharie[17], Robin L. Haynes [1]*

1 Department of Pathology, Boston Children's Hospital, Harvard School of Medicine, Boston, Massachusetts, United States of America, 2 Neuropathology Brain Bank and Brain Injury Research Center, Department of Rehabilitation and Human Performance, Icahn School of Medicine at Mount Sinai, New York, New York, United States of America, 3 RTI International, Research Triangle Park, Durham, North Carolina, United States of America, 4 Department of Obstetrics and Gynecology, Faculty of Medicine and Health Science, Stellenbosch University, Cape Town, South Africa, 5 Carelon Research, Newton, Massachusetts, United States of America, 6 Avera Research Institute, Sioux Falls, South Dakota, United States of America, 7 Anatomic Pathology, Texas Children's Hospital, Baylor College of Medicine, Houston, Texas United States of America, 8 Forensic Medicine, Forensic and Analytical Science Service, New South Wales Health Pathology, Lidcombe, Australia, 9 Department of Pathobiology and Integrative Biomedical Sciences, College of Veterinary Medicine, Dalton Cardiovascular Research Center, University of Missouri, Columbia, Missouri, United States of America, 10 Latrobe Regional Health, Traralgon, Victoria, Australia, 11 Department of Psychiatry and Pediatrics, Columbia University Medical Center, New York State Psychiatric Institute, New York, New York, United States of America, 12 Epidemiology, Statistics, and Population Sciences, National Institute on Deafness and Other Communication Disorders, NIH, Bethesda, Maryland, United States of America, 13 Department of Molecular and Systems Biology, Geisel School of Medicine at Dartmouth, Hanover, New Hampshire, United States of America, 14 MaineGeneral Medical Center, Augusta, Maine, United States of America, 15 Department of Pathology, University of South Dakota Sanford School of Medicine, Vermillion, South Dakota, United States of America, 16 Department of Pathology, Massachusetts General Hospital, Boston, Massachusetts United States of America, 17 Division of Anatomical Pathology, Department of Pathology, Faculty of Medicine and Health Science, Stellenbosch University, Cape Town, South Africa, 18 Department of Pathology, University of North Dakota, School of Medicine and Health Sciences, Grand Forks, North Dakota, United States of America, 19 University of Stellenbosch, Western Cape, South Africa, 20 National Health Laboratory Services, Nelson Mandela Bay, South Africa

* robin.haynes@childrens.harvard.edu

## Abstract

The Sudden Infant Death Syndrome (SIDS) is a major global health problem, with increased risk among socioeconomically disadvantaged populations. We propose SIDS, or a subset, is due to a defect in the brainstem serotonin system mediating cardiorespiratory integration and arousal. This defect impinges on homeostasis during a critical developmental period in infancy, especially in populations experiencing maternal and infantile stress, resulting in sleep-related sudden death. In the socially disadvantaged cohort of the *prospective* Safe Passage Study from Cape Town, South

**Data availability statement:** De-identified data from the Safe Passage Study is available through NICHD's Data and Specimen Hub (DASH). All case demographic and exposure data are available on DASH. Elliott, Amy (2025). A Prospective Study on the Role of Prenatal Alcohol Exposure in SIDS and Stillbirth (Version 1). NICHD Data and Specimen Hub. https://doi.org/10.57982/sv8c-4y07. Ligand receptor binding data are available as S2 File. The tribal data used in this study is restricted access per the requirements of participating tribal nations and the Indian Health Service IRB. Avera Health maintains the data on a secure server and people can contact Dr. Christine Hockett (Christine.hockett@avera.org) to learn the process for gaining tribal approval and necessary regulatory approvals to gain access.

**Funding:** The research reported in this publication was supported by National Institutes of Health (NIH) grants U01HD045935 (MEN, JA, AJE, LLN, BBR, MAS ), U01HD055155 (WPF), U01HD045991 (HCK, RDF, KGB, JRD, DSP), and U01AA016501 (LB, EHB, JAC, HJO) funded by the National Institute on Alcohol Abuse and Alcoholism (https://www.niaaa.nih.gov/), Eunice Kennedy Shriver National Institute of Child Health and Human Development (https://www.nichd.nih.gov/), and the National Institute on Deafness and Other Communication Disorders (https://www.nidcd.nih.gov/). There was no additional external funding received for this study. The work was performed in the context of the research network Prenatal Alcohol, SIDS, and Stillbirth (PASS) Network. This Network included affiliates of the NIH who participated in the overall PASS study design, decision to publish, and preparation of the manuscript.

**Competing interests:** The authors have declared that no competing interests exist.

Africa, and the Northern Plains of the United States, we tested the hypothesis that: 1) serotonin (5-HT) receptor 1A binding is reduced within the brainstem of SIDS infants compared to controls; and 2) reduced $5\text{-HT}_{1A}$ binding in SIDS is associated with maternal drinking and/or smoking during pregnancy. Using receptor ligand autoradiography for the $5\text{-HT}_{1A}$ receptor, $^3\text{H-8-OH-DPAT}$ binding was measured in brainstem nuclei in infants dying of SIDS (n = 14) and controls dying of known causes (n = 10). We found a brainstem serotonin defect in SIDS infants, that is strongly driven by preterm birth, and that likely underlies the pathogenesis of sleep-related sudden death in response to homeostatic stress. The findings replicate studies of US low-to-middle income SIDS cohorts, with key differences related to prematurity, including increased $5\text{-HT}_{1A}$ binding in premature SIDS compared to premature controls. The relationship of the serotonin defect to prenatal smoking and drinking is unclear, owing to the high exposure rates in SIDS cases *and* controls. SIDS was significantly associated with lack of a phone (proxy for poverty) (p = 0.024) and overcrowded housing (p = 0.047). These data support the concept of a serotonin defect in brainstem nuclei mediating cardiorespiratory control and arousal in SIDS infants. Maternal and/or fetal stress, along with premature birth, may underlie a deflection of normal development of the serotonergic system.

## Introduction

The Sudden Infant Death Syndrome (SIDS) is defined as the sudden and unexpected death of an apparently healthy infant in the first postnatal year, which remains unexplained after a complete anatomic autopsy and forensic investigation [1]. SIDS is a primary cause of infant death (24.16/100,000 deaths per live births worldwide [2]) and the leading cause of postneonatal death in the United States [3]. SIDS is indiscriminate, affecting infants of both sexes from all racial, ethnic, and socioeconomic backgrounds, though the disorder is disproportionately represented in socially disadvantaged populations [4–7]. SIDS is associated with sleep periods and sleep environments conducive to life-threatening asphyxia (e.g., prone sleep position and bed sharing) [3,8,9]. These observations underlie a leading hypothesis that at least a subset of SIDS is due to the failure of protective cardiorespiratory responses to asphyxia and cardiovascular collapse during sleep [10–20]. Other risk factors for SIDS include maternal drinking and/or smoking during pregnancy, low birth weight, maternal anemia during pregnancy, and prematurity, all suggestive of a suboptimal intrauterine environment [21–27]. These observations suggest the possibility that SIDS, although occurring during a critical *postnatal* period, has a *prenatal* origin.

Over the last three decades, we and others have identified abnormalities within the serotonergic system – i.e., serotonin (5-HT) neurons and their targets expressing 5-HT receptors – of the medulla oblongata (lower brainstem) of a subset of SIDS infants [18,28–38]. We classified these SIDS deaths as "serotonopathies", a term encompassing disorders within the serotonergic system. 5-HT, via interaction with

5-HT receptors differentially expressed across regions of the medulla, is important for cardiorespiratory homeostasis in sleep including arousal, hypercapnic ventilatory responses, airway patency, and autoresuscitation, the latter a protective response that preserves life in the face of severe brain tissue hypoxia [39–44]. In this SIDS subset, pathophysiological events that lead to sudden death likely involve defects within both serotonergic source nuclei (i.e., those that synthesize 5-HT) and medullary target nuclei (those that express 5-HT receptors but do not synthesize 5-HT). Within the medulla, we have demonstrated that SIDS is associated with reduced 5-HT at source nuclei [29] and reduced 5-HT$_{1A}$ and 5-HT$_{2A/C}$ receptor binding in several target nuclei involved in cardiorespiratory control and/or arousal [18,28,30,32], findings that have been replicated in multiple independent cohorts over the last two decades [35–38,45]. Experiments in animal models provide the important biological plausibility that 5-HT defects identified in human tissue compromise cardiorespiratory responses to asphyxia and, therefore, have a direct, causative role in SIDS [16,17,42,46–50].

Prenatal exposure to alcohol and tobacco smoke are key SIDS risk factors that have received increasing attention in public health campaigns [51,52]. These risk factors are considered modifiable by appropriate behavioral interventions in drinkers and smokers, including in pregnant women [53,54]. SIDS risk increased ~12-fold when mothers in the prospective Safe Passage Study (SPS) smoked and drank during pregnancy beyond the first trimester [23]. In a retrospective study of the American Indians of the Northern Plains [55], the risk of SIDS was increased ~6-fold when the mother drank alcohol in the periconceptual period. In retrospective studies in American Indians and non-indigenous persons, we found altered nicotinic receptor binding in the pons of infants of mothers who smoked during pregnancy [56,57]. The increased risk of SIDS was associated with reduced serotonergic receptor binding in SIDS infants within vital cardiorespiratory nuclei in the medulla [33]. Moreover, reduced serotonergic binding at the ventral medullary surface – a region contributing to central $CO_2$ chemosensitivity – was associated with prenatal exposure to cigarette smoke (p = 0.011) and alcohol consumption (p = 0.075) [33].

The finding that alcohol and cigarette smoke influenced the brainstem serotonergic system prompted the initiation of the SPS, an international, *prospective*, observational, multi-center study with five clinical sites in Cape Town and two in the Northern Plains [58]. The study populations in the SPS were selected for high rates of SIDS and exposure to maternal drinking and smoking during pregnancy. While maternal drinking and smoking have documented effects on fetal development in general and specific effects on the brainstem serotonergic system [59–63], the mechanisms by which these exposures increase the risk for SIDS remain unclear. The overriding hypothesis of the SPS was that prenatal alcohol and/or cigarette smoke exposure, modified by other environmental factors, were associated with serotonergic abnormalities in the medullary 5-HT system in SIDS. We refined this hypothesis in the current study and proposed that 5-HT$_{1A}$ receptor binding would be low in the brainstem of infants who died of SIDS in the SPS cohort, as found in previous studies of San Diego SIDS cohorts [28,29]. We further hypothesized that low 5-HT$_{1A}$ binding would be most apparent in SIDS infants born to mothers who smoked and drank during pregnancy, an idea not tested previously in the San Diego cohorts due to the lack of rigorously acquired, quantitative, prospective data pertaining to behavioral risks—a strength of the SPS. Of note, the current SPS cohort was also associated with low socioeconomic status [23,58]. We studied the binding of the radioligand ³H-8-OH-DPAT to 5-HT$_{1A}$ receptors, a proxy marker of 5-HT$_{1A}$ activity, in the brainstems of infants dying of SIDS and controls with known causes of death accrued from the SPS.

## Materials and methods

### Design of the safe passage study

The overall recruitment period was 08/01/2007 - 01/30/2015. For the South African site, recruitment was from 08/06/2007 to 01/13/2015. For the Northern Plains site, recruitment was from 08/01/2007 to 01/30/2015. The hypotheses, specific aims, common protocol, enrollment, shipping, compliance, and specimen donation of the SPS have been described in detail [58], as has the approach to autopsy consent in socioeconomically disadvantaged populations [64]. Clinical sites were selected based upon known high rates of maternal drinking and smoking during pregnancy and known high rates of SIDS in the population; however,

all women from the catchment areas presenting for care at these sites were eligible to participate. Participants included (1) Caucasian and American Indian mothers of the Northern Plains, and (2) mixed ancestry mothers (Cape Coloured) of the Western Cape, South Africa. Screening and enrollment occurred at the prenatal clinics affiliated with each clinical site between 6 weeks gestation and term. Gestational age at enrollment was determined during the first prenatal visit using standard clinical practices at each study center – ultrasound in South Africa, and a combination of clinical examination, ultrasound, and last menstrual period in the Northern Plains [65]. The maternal and fetus/infant dyads were followed during pregnancy and after delivery until infants were 1 year of age, i.e., the risk period of SIDS. Detailed information regarding quantity, frequency, and timing of substance use was self-reported up to 4 times during pregnancy (at recruitment, 20–24, 28–32, and 34+gestational weeks) and at 1-month post-delivery. Upon demise, an autopsy was routinely ordered by the coroner/medical examiner, after which the family was approached for consent by a research team member for the donation of tissue for research purposes. After written informed consent was obtained from either parent of the deceased infant, brain portions were frozen and shipped on dry ice to the Developmental Brain and Pathology Center (DBPC), Department of Pathology, Boston Children's Hospital, the centralized laboratory for research analysis [58]. An external advisory committee provided oversight of the SPS. The institutional boards of the local hospitals at which the infants were autopsied as well as Boston Children's Hospital, approved the use of brain tissues in the Safe Passage Study. If the study participant (mother of the deceased infant) was a minor mother, the parent or guardian of the minor mother provided written consent for her participation in the research.

## Ethics statement

The consents for all parts of the Safe Passage Study were in writing and witnessed by research staff. If there was a fetal or infant demise, there was an additional written consent at that point in time for donation of autopsy tissue for research purposes. The original Safe Passage Study protocol and all subsequent modifications and addendums were approved by the Ethics Committee of each individual site, including the Health Research Ethics Committee of Stellenbosch University, Institutional Review Board (IRB) of Avera, and IRB of Boston Children's Hospital.

## Clinical database

SIDS, as defined above [1], included deaths that might otherwise have been classified as undetermined, including infants dying in unsafe sleep conditions but without evidence of mechanical asphyxia or suffocation by overlying. Known causes of death (KCOD) controls were defined as infants whose cause of death was determined after review of all available information, including from an autopsy [58]. Details of the adjudication processes were described previously [58]. Prospective collection of those infants who died and had an autopsy in the SPS, the demise cohort, included 28 SIDS and 38 control-cases who died after discharge from the hospital (postdischarge known cause of death controls [PostKCOD]). Of these, 14 SIDS infants and 28 PostKCOD infants did not have brain tissue available for neurochemistry either due to a lack of consent for autopsy research or due to technical issues related to the quality of tissue. Of the 14 SIDS and 10 PostKCOD controls available and of suitable quality for autoradiography, 9 (64%) SIDS had rostral pons available but with varying availability of regions of interest within the pons (n=6–9). Four PostKCOD controls (40%) had rostral pons available. There were 45 KCOD controls who died after delivery but prior to leaving the hospital (predischarge KCOD; PreKCOD), from which 12 brains were available and of suitable quality to provide baseline developmental data for $^3$H-8-OH-DPAT binding. The comparative analysis of socioeconomic variables was investigated only in the South Africa prospective demise SIDS infants and controls. The SPS design group prespecified that any infant who died prior to discharge from the hospital should not be defined as a SIDS death. However, the brain development of PreKCOD infants is on a continuum with the brain development of PostKCOD infants and SIDS infants. We therefore performed a complete analysis of this continuum of PreKCOD and PostKCOD controls to determine whether the 5-HT$_{1A}$ binding differences between SIDS and control infants were consistent regardless of the discharge status of the controls.

## 5-HT$_{1A}$ receptor binding and generation of brainstem autoradiograms

Brainstem processing was performed as previously described in the San Diego Cohort [28,29] and the SPS cohort [66,67]. The autoradiography procedures for determination of ³H-8-OH-DPAT (³H-8-hydroxy-2-[di-*N*-propylamino]-tetralin) binding to 5-HT$_{1A}$ receptors was performed on 20µ thick sections according to previously described protocols [28,29]. The use of different radioligands for 5-HT receptors in the 2002 Northern Plains study and the SPS precluded precise comparison of the binding between the two studies. Total 5-HT$_{1A}$ receptor binding was determined by incubation of tissue sections in 4-nM ³H-8-OH-DPAT (Revvity, Waltham, MA). Nonspecific binding was measured using 10 µM of unlabeled 5-HT. For each specimen, receptor binding density was analyzed in 13 medullary nuclei (all nuclei were not available in all cases) at 2 defined levels of the brainstem (2 autoradiograms for each nucleus) according to previously published methods [28]. Quantitative densitometry of autoradiograms was performed blinded to diagnosis or age using an MCID 5+ imaging system (Imaging Research Inc, St Catharines, Ontario). Autoradiography binding was performed in batches, with cases blinded to diagnosis, randomly distributed across the different batches. The same autoradiography standard was used across the different batches.

## Analysis of 5-HT$_{1A}$ binding in homeostatic brainstem sites

The human brainstem sites measured in this study (Fig 1) were defined with reference to the Olszewski and Baxter human brainstem atlas [68] and confirmed with Paxinos and Huang human brainstem atlas [69]. The functions of the sampled nuclei were summarized previously [32]. For each case, 5-HT$_{1A}$ binding in the brainstem was measured at 3 levels. The mid-medulla at the level of nucleus of Roller included the nucleus of the solitary tract (NTS) (all visceral sensory inputs of the autonomic nervous system and sympathetic autonomic system integration), the hypoglossal nucleus (HG) (airway patency, especially during sleep), dorsal motor nucleus of the vagus (DMX) (preganglionic vagal outflow of the parasympathetic autonomic nervous system, cardiac neurons), centralis (CEN) (caudal central reticular formation, cardiorespiratory integration), principal inferior olive (PIO), medial accessory olive (MAO) (the olivocerebellar network, including blood pressure recovery), principal spinalis trigeminalis (S5) (pain and temperature of cranial nerve five), the

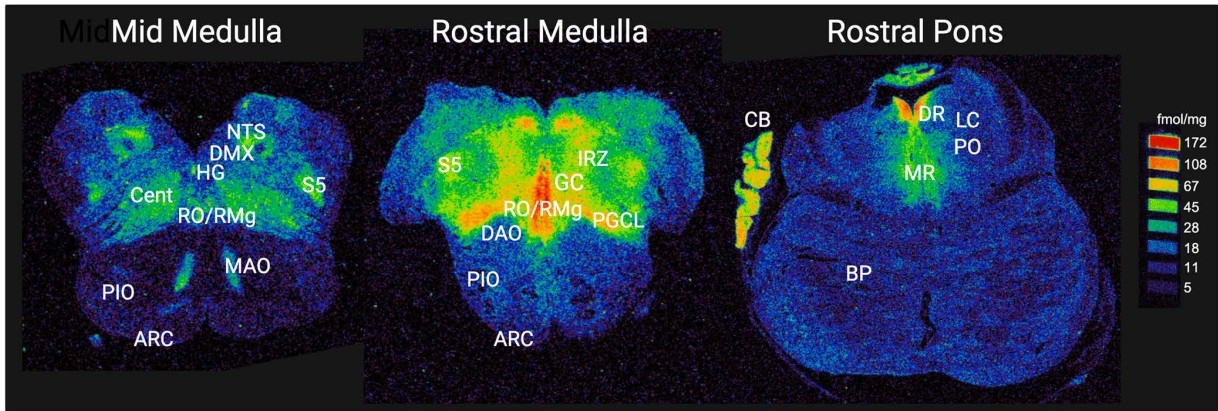

**Fig 1. Representative distribution of 5-HT$_{1A}$ binding in the pons and medulla.** Autoradiograph images display the distribution pattern of 5-HT$_{1A}$ binding across different nuclei of the medulla and rostral pons. Receptor binding in fmol/mg is indicated with color according to the scale. Measured nuclei of the mid medulla are as follows: RO/RMg, raphe obscurus/raphe magnus; HG, hypoglossal nucleus; DMX, dorsal motor nucleus of the vagus; NTS, nucleus of the solitary tract; CEN, centralis; S5, spinal 5; MAO, medial accessory olive; PIO, principal inferior olive; ARC, arcuate nucleus. Measured nuclei of the rostral medulla are as follows: RO/RMg, raphe obscurus/raphe magnus; GC, gigantocellularis; PGCL, paragigantocellularis lateralis; IRZ, intermediate reticular zone; PIO, principal inferior olive; DAO, dorsal accessory olive; ARC, arcuate nucleus. Measured nuclei of the rostral pons are as follows: MR, median raphe; DR, dorsal raphe; LC, locus coeruleus; PO, nucleus pontis oralis; BP, basis pontis. Cerebellar tissue (CB, rostral pons section only) displays 5-HT$_{1A}$ binding but was not measured.

raphe obscurus/raphe magnus (RO/RMg) (cardiorespiratory integration and arousal responses), and the arcuate nucleus (ARC) (putative homologue of ventral chemosensory zone at the medullary surface). The rostral medulla at the level of the nucleus prepositus included the gigantocellularis (GC) (putative homologue of preBotzinger complex), paragigantocellularis lateralis (PGCL), intermediate reticular zone (IRZ), RO/RMg, S5, DAO, PIO, and ARC. The rostral pons at the level of nucleus parabrachialis lateralis, included the locus coeruleus (LC) (major source neurons of the noradrenergic ascending arousal network), the pontis oralis (PO) (part of source neurons of the cholinergic ascending arousal system), median raphe (MR) and dorsal raphe (DR) (components of the rostral 5-HT ascending arousal network), and the basis pontis (BP) (Fig 1). Except for the midline nuclei (RO and MR), 5-HT$_{1A}$ binding was measured from both left and right sides of the section, and the mean binding of the two sides was calculated to determine the final value of binding in fmol/mg tissue. Receptor binding data for 5-HT1A are provided in S2 File.

### Prospective collection of prenatal exposure data

The SPS used a modified Timeline Follow Back method to collect exposures related to maternal smoking and drinking during pregnancy [58]. At each visit during and after pregnancy, the mother was asked about the last date of use (separately for alcohol and smoking). For data relating to alcohol, they were asked about consumption for ± 15 days around last menstrual period, as well as the 30 days prior to the last drinking day since their last research visit. For smoking, they were asked about the frequency of smoking and number of cigarettes on a typical day for the 30 days prior to the last date of use since their last research visit. To estimate the total number of drinks consumed during pregnancy, and the average cigarettes smoked per week during pregnancy, missing values were using the k-Nearest Neighbor (kNN) method [70,71].

### Data collection and variable description

Maternal demographics assessed included age, education, housing type ("good" defined as an apartment or house with a crowding index [ratio of number of rooms to number of occupants] of <1.5; "poor" defined as small model "play" house, informal shack, shelter, or trailer, often with no running water, and with crowding index of ≥1.5) [72], history of loss by SIDS, and delivery type. Maternal demographics also included an anxiety trait score based on The Spielberger State-Trait Anxiety Inventory (STAI) [73] and the Edinburgh Postnatal Depression Scale [74]. Infant demographics assessed included birth weight and length, gestational age at birth, postnatal age at death, gender, and race. Autopsy findings assessed included post-mortem interval, body weight at autopsy, and brain weight at autopsy. Maternal use of alcohol and smoking during pregnancy was assessed as binary values (used during pregnancy or not) and as continuous values (total number of drinks during pregnancy and average number of cigarettes per week). Maternal use of alcohol and smoking by trimester was accessed as continuous values (number of drinks by trimester and average cigarettes per week). We also recorded the sleep position (supine vs. prone) that the infant was last placed.

### Statistical analysis

Differences between cause of death (SIDS vs KCOD controls) in demographic characteristics, maternal substance use during pregnancy, and relevant autopsy and clinical findings were tested by ANOVA for continuous variables and Chi-square testing or Fisher's exact test for categorical variables. Linear regression analysis assessed the effect of development [postconceptional age (PCA)] on 5-HT$_{1A}$ binding in the 12 preKCOD- and 10 PostKCOD-control infants. Linear regression modelling was used to analyze differences in mean 5-HT$_{1A}$ binding values by case diagnosis, controlling for PCA. Consistent with prior findings in several cohorts of SIDS cases, and specified *a priori* in the SPS study, the interaction of diagnosis and PCA was included in this modelling when the interaction was statistically significant. When the interaction was not significant, it was dropped from the final model but reported for completeness; when the interaction was significant, means by diagnosis are not reported, as the significant interaction indicates that the means vary by age. A post-hoc, exploratory linear regression assessed differences in mean 5-HT$_{1A}$ binding values by diagnosis, stratified

by preterm vs. term, controlling for the effect of PCA. Although testing for interactions with such small sample sizes is generally not advisable, interactions between diagnosis and PCA were tested due to strong and consistent prior findings in SIDS. Finally, linear regression models analyzed differences in mean 5-HT$_{1A}$ binding values by number of drinks per pregnancy and average cigarettes per week during pregnancy (separate models), controlling for PCA. Sample sizes were too small to assess diagnosis and exposure on 5-HT$_{1A}$ binding at the same time. However, in post-hoc exploratory analysis, t-tests were used to assess differences in exposures by diagnosis and by prematurity. Statistically, there was no effect of postmortem interval (PMI) on binding, as determined by linear regression of PMI with binding levels. Therefore, PMI was not controlled for in any analyses. Analyses were conducted using SAS 9.4 and SAS EG 7.15. There were no outliers excluded from statistical analysis and complete case analysis was performed. In all analyses, a p-value ≤0.05 was accepted for statistical significance. No formal adjustment for multiple testing was performed, but consistency of results across multiple related outcomes is emphasized.

## Results

### Clinicopathological information

The demise cohort for $^3$H-8-OH-DPAT analyses included SIDS (n = 14), controls who died after discharge from the hospital (PostKCOD; n = 10), and controls who died before discharge from the hospital (PreKCOD; n = 12). The causes of death for the control groups are given in Table 1 and represent a range of causes and complicatons. Selected demographic data for all KCOD and SIDS (South Africa and Northern Plains) infants are summarized in Table 2. PreKCOD cases (n = 12) ranged in age of death from 24 postconceptional (PC) weeks to 42 PC weeks (mean = 32.6 weeks) and were included specifically to examine developmental changes in receptor binding at this relatively young age (Table 2). Sixty-seven percent of PreKCOD cases were male (n = 8), and 83% (n = 10) were South African mixed race, and the other 17% were American Indian (n = 1) or Caucasian (n = 1). When the SIDS cohort was compared to the PostKCOD cases only, there was no significant difference in mean gestational age (GA), postconceptional age (PCA), postmortem interval (PMI), birth weight, sex, incidence of premature birth, autopsy body or brain weight between SIDS and PostKCOD infants (Table 2).

Within the whole cohort (South African and Northern Plains), the incidence of maternal smoking during pregnancy was 100% (14/14) in the SIDS group (Table 3) and ranged from an average of 0.1 cigarettes per week to 62.3 [median of 20.2] (S1 Table 1). This was not statistically different from PostKCOD cases in whom the incidence of smoking was 90% (9/10) (Table 3) and ranged from an average of 0 cigarettes per week to 58.6 [median of 28.7] (S1 Table 1). The incidence of maternal drinking during pregnancy was 57% (8/14) in the SIDS group (Table 3) and ranged from 0 drinks to 210.8 drinks during the pregnancy [median of 8.8 drinks/pregnancy] (S1 Table 1). This range of alcohol consumption was not significantly different from PostKCOD cases in whom the incidence of drinking was 70% (7/10) (Table 3) and ranged from 0 drinks to 102.5 drinks during pregnancy [median of 8.3 drinks/pregnancy] (S1 Table 1). There was no statistical difference between SIDS and PostKCOD cases when exposure was analyzed by trimesters (S1 Table 1). There was no significant difference in alcohol and cigarette exposure within the following categories: SIDS, preterm vs. term infants; PostKCOD, preterm vs. term infants; preterm infants, SIDS vs. controls; and term infants, SIDS vs. controls. (S1 Tables 2–5). The sleep position of the infant when found at the time of death was documented on a subset of SIDS (n = 13) and PostKCOD controls (n = 2). 9/13 (69%) SIDS cases were found on their side and 4/13 (31%) SIDS cases were found on their stomach. One PostKCOD was found sleeping on the back and one PostKCOD was found sleeping on the stomach.

### Characteristics of maternal socioeconomic status and mental health

We analyzed key environmental factors including phone ownership, household income, quality of housing, and the degree of crowding within the home. In addition, we analyzed indices of maternal anxiety and depression. Despite no overall difference in household income between SIDS and control mothers, a significantly higher proportion of mothers of control infants owned a phone compared to SIDS mothers (94% vs. 62%; p = 0.024; Table 4), suggesting financial constraints for

**Table 1.** Causes of death in pre- and post-discharge known cause of death (KCOD) cases: Combined South Africa and Northern Plains cohorts.

| Case | Pre- or Post-discharge | GA (wks) | PNA (wks) | PCA (wks) | Cause of Death |
|------|------------------------|----------|-----------|-----------|----------------|
| 1 | Pre-discharge | 24.1 | 0.1 | 24.2 | Hyaline membrane disease, complications of prematurity |
| 2 | Pre-discharge | 25.7 | 0.1 | 25.8 | Complications of prematurity |
| 3 | Pre-discharge | 27.0 | <0.1 | 27.0 | Hyaline membrane disease, chorioamnionitis and placental abruption |
| 4 | Pre-discharge | 27.0 | 0.6 | 27.6 | Preeclampsia and prematurity |
| 5 | Pre-discharge | 30.3 | 1.4 | 31.7 | Omphalocele, peritonitis, sepsis |
| 6 | Pre-discharge | 32.6 | 0.1 | 32.7 | Pulmonary hemorrhage |
| 7 | Pre-discharge | 32.9 | 0.1 | 33.0 | Fetal head trauma due to motor vehicle accident |
| 8 | Pre-discharge | 31.7 | 1.6 | 33.3 | Klebsiella pneumonia, necrotizing enterocolitis, jaundice |
| 9 | Pre-discharge | 37.0 | <0.1 | 37.0 | Intrauterine growth restriction |
| 10 | Pre-discharge | 37.5 | <0.1 | 37.5 | Pulmonary hypoplasia, multicystic dysplastic kidney disease (Potter's sequence) |
| 11 | Pre-discharge | 39.2 | 0.75 | 40 | Chorioamnionitis, ascending infection, sepsis, pneumonia, meconium aspiration |
| 12 | Pre-discharge | 41.3 | 0.6 | 41.9 | Meconium aspiration, severe bronchopneumonia, perinatal asphyxia |
| 13 | Post-discharge | 27.3 | 9.4 | 36.7 | Respiratory infection |
| 14 | Post-discharge | 35.6 | 2.1 | 38.0 | Respiratory infection |
| 15 | Post-discharge | 27.6 | 12.9 | 40.5 | CNS infection |
| 16 | Post-discharge | 38.9 | 1.6 | 40.5 | Congenital defects |
| 17 | Post-discharge | 32.1 | 12.1 | 44.2 | Renal; tubulo-interstitial nephritis |
| 18 | Post-discharge | 36.0 | 9.0 | 45.0 | Respiratory infection |
| 19 | Post-discharge | 40.0 | 11.7 | 51.7 | Respiratory infection |
| 20 | Post-discharge | 38.3 | 14.1 | 52.4 | CNS infection |
| 21 | Post-discharge | 39.6 | 22.7 | 62.3 | Gastrointestinal infection |
| 22 | Post-discharge | 38.6 | 25.9 | 64.4 | Respiratory infection |

Legend. Pre-discharge cases are infants who died in the hospital, i.e., prior to release. Post-discharge cases are infants that died after discharge from the hospital. Abbreviation: KCOD, known cause of death; CNS, central nervous system; N, number; GA, gestational age; PNA, postnatal age; PCA, postconceptional age; wks, weeks.

SIDS mothers. The disparity between SIDS and control mothers regarding phone ownership was driven by those infants who were born prematurely; 43% of premature SIDS mothers owned a phone compared to 91% of premature control mothers (p = 0.025; Table 4), while there was no difference in phone ownership between mothers of SIDS and control infants who were born at term.

Forty-six percent of mothers of infants dying with the classification of SIDS lived in housing deemed as "poor", compared to 18% of mothers of control infants (p = 0.091; Table 4). Notably, 86% of SIDS infants born prematurely lived in poor housing while there were no SIDS infants born at term who lived in such conditions (p = 0.001; Table 4). As with phone ownership, the higher incidence of poor housing for mothers of future SIDS infants was driven largely by the housing of infants born prematurely (control: 18%; SIDS: 86%; p = 0.003; Table 4). The homes of SIDS infants were overcrowded relative to control infants (p = 0.047; Table 4) an effect that tended to be driven by the housing of infants born prematurely (p = 0.085; Table 4).

In addition to the environmental factors described above we also examined the potential impact of these factors on the mental health of mothers of SIDS and control infants, as determined by anxiety and standard depression measures [58]. Mothers of SIDS infants tended to have higher rates of anxiety (p = 0.072) and depression (p = 0.076) compared to control mothers (Table 4).

**Table 2. Infant demographics: Combined South Africa and Northern Plains cohorts.**

| | PreKCOD | | PostKCOD | | SIDS | | p-values |
|---|---|---|---|---|---|---|---|
| | n | Mean±STD [or n (%)] | n | Mean±STD [or n (%)] | n | Mean±STD [or n (%)] | SIDS vs. Post KCOD |
| Post conceptional age (weeks) | 12 | 32.6±5.6 | 10 | 47.6±9.8 | 14 | 49.7±11.2 | 0.636 |
| Gestational age (weeks) | 12 | 32.2±5.6 | 10 | 35.4±4.8 | 14 | 36.6±3.5 | 0.481 |
| Post-mortem Interval (hours) | 12 | 41.5±32.7 | 10 | 47.6±37.1 | 14 | 42.5±23.4 | 0.686 |
| Male | 12 | [8 (67)] | 10 | [6 (60)] | 14 | [6 (43)] | 0.408 |
| Race | 12 | | 10 | | 14 | | 0.700 |
| American Indian | | [1 (8)] | | [1(10)] | | [0 (0)] | |
| South African Mixed Race | | [10 (83)] | | [8(80)] | | [13 (93)] | |
| Caucasian | | [1 (8)] | | [1(10)] | | [1 (7)] | |
| Premature Birth | 12 | [8 (67)] | 10 | [5 (50)] | 14 | [7 (50)] | 1.000 |
| Birth Weight (grams) | 12 | 1661.5±927.5 | 10 | 2535.0±1061.0 | 14 | 2472.9±756.7 | 0.868 |
| Birth Length (centimeters) | 4 | 37.8±9.1 | 8 | 47.8±5.7 | 12 | 47.5±4.0 | 0.906 |
| Autopsy body weight (grams) | 12 | 2216.1±1651.1 | 9 | 3672.4±1870.0 | 14 | 4333.1±1829.1 | 0.411 |
| Autopsy brain weight (grams) | 11 | 236.4±130.3 | 6 | 507.8±238.1 | 12 | 582.7±253.6 | 0.555 |

Legend. P-values compare SIDS to PostKCOD. N represents the number of cases with available demographic information. Abbreviations. SIDS, sudden infant death syndrome; STD, standard deviation; PreKCOD, predischarge known cause of death; PostKCOD, postdischarge known cause of death.

**Table 3. Incidence of exposure during pregnancy in the SPS cohort: Combined South Africa and Northern Plains cohorts.**

| | PreKCOD | | PostKCOD | | SIDS | | p-value |
|---|---|---|---|---|---|---|---|
| | n | n (%) [or Median] | n | n (%) [or Median] | n | n (%) [or Median] | |
| Alcohol during pregnancy (yes/no) | 10 | 7 (70) | 10 | 7 (70) | 14 | 8 (57) | 0.678 |
| N drinks in pregnancy | 12 | [4.4] | 10 | [8.3] | 14 | [8.8] | 0.700 |
| Smoking during pregnancy (yes/no) | 10 | 10 (100) | 10 | 9 (90) | 14 | 14 (100) | 0.417 |
| Average cigarettes per week | 8 | [21.4] | 8 | [28.7] | 13 | [20.2] | 1.000 |

**Legend.** Abbreviations. KCOD, Known cause of death; STD, standard deviation; N, number; n, number.

## Strategy

We approached the analysis of 5-HT$_{1A}$ receptor binding in the demise cohort of the SPS in a series of steps, examining in order: 1) the developmental profile of controls spanning postconceptional age from mid-gestation to the end of the first postnatal year; 2) the main effects of diagnosis and interactions (age-effects) compared between the SIDS cases vs. age-related control infants (PostKCOD; see definitions above), and 3) a comparison of preterm SIDS cases vs. preterm controls and separately, term SIDS vs. term controls. The third approach was not specified *a priori*, but was based on our recognition that the demise cohort had a large proportion of premature infants (50% of both SIDS and PostKCOD controls), providing the opportunity for further study. The results in each of the three steps are presented below.

## Developmental pattern of 5-HT$_{1A}$ receptor binding throughout the first postnatal year

We measured 5-HT$_{1A}$ binding within 5-HT source and target nuclei (see definitions above) in PreKCOD (n = 12) and PostKCOD (n = 10) controls from 24 to 64 PC weeks. The purpose of the analysis was to assess the possibility that 5-HT$_{1A}$ binding changed across development. While we acknowledge that the PreKCOD infants had serious health complications (Table 1), their inclusion strengthens the interpretation of age-related findings. Considering PreKCOD and PostKCOD

**Table 4. Maternal socioeconomic characteristics: South African cohort only.**

| Variable | SIDS (n=13) All KCOD (n=18) Mean or n (%) | p-value | SIDS (n=13) PostKCOD (n=8) Mean or n (%) | p-value | SIDS Premature (7) Term (6) Mean or n (%) | p-value | All KCOD Premature (12) Term (6) Mean or n (%) | p-value | Premature Infants SIDS (7) All KCOD (12) Mean or n (%) | p-value | Term Infants SIDS (6) All KCOD (6) Mean or n (%) | p-value | All KCOD PreKCOD (10) PostKCOD(8) Mean or n (%) | p-value |
|---|---|---|---|---|---|---|---|---|---|---|---|---|---|---|
| **Environmental** | | | | | | | | | | | | | | |
| Phone possession | S 8 (62) K 16 (94) | **0.024** | S 8 (62) K 7 (100) | **0.023** | P 3 (43) T 5 (83) | 0.135 | P 10 (91) T 6 (100) | 0.342 | S 3 (43) K 10 (91) | **0.025** | S 5 (83) K 6 (100) | 0.224 | Post 7 (100) Pre 9 (90) | 0.293 |
| Poor housing* | S 6 (46) K 3 (18) | *0.091* | S 6 (46) K 1 (14) | 0.137 | P 6 (86) T 0 (0) | **0.001** | P 2 (18) T 1 (17) | 0.937 | S 6 (86) K 2 (18) | **0.003** | S 0 (0) K 1 (17) | 0.224 | Post 1 (14) Pre 2 (20) | 0.759 |
| Income (Rands) | S 476 K 644 | 0.256 | S 476 K 568 | 0.576 | P 527 T 401 | 0.496 | P 819 T 383 | **0.031** | S 527 K 819 | 0.149 | S 401 K 383 | 0.908 | Post 568 Pre 710 | 0.510 |
| Crowding index* | S 2.2. K 1.4 | **0.047** | S 2.2. K 1.7 | 0.278 | P 2.2 T 2.3 | 0.857 | P 1.4 T 1.4 | 0.969 | S 2.2. K 1.4 | *0.085* | S 2.3. K 1.4 | 0.238 | Post 1.7 Pre 1.3 | 0.207 |
| **Maternal** | | | | | | | | | | | | | | |
| Anxiety trait >40 | S 7 (54) K 5 (29) | 0.175 | S 7 (54) K 1 (14) | *0.072* | P 3 (43) T 4 (67) | 0.388 | P 3 (27) T 2 (33) | 0.794 | S 3 (43) K 3 (27) | 0.496 | S 4 (67) K 2 (33) | 0.244 | Post 1 (14) Pre 4 (40) | 0.238 |
| Edinburgh Total Score | S 14 K 10 | *0.076* | S 14 K 10 | 0.141 | P 13 T 16 | 0.271 | P 10 T 10 | 0.873 | S 13 K 10 | 0.487 | S 16 K 10 | *0.079* | Post 10 Pre 10 | 0.986 |

**Legend.** Abbreviations. KCOD, known cause of death; Post, postdischarge; Pre, predischarge; S, SIDS; K, Known; n, number; NS, not significant P, premature; T, term; Bold, significant p-value < 0.05; Italic, p-value trend towards significance.*, see text for definition of term.

infants together, the highest binding was in the ROB/RMg, a 5-HT source nucleus containing 5-HT-synthesizing neurons and critical for cardiorespiratory reflex integration (Table 5). In this combined group of infants not classified as SIDS, 5-HT$_{1A}$ binding within 7 nuclei sampled, including 5-HT source nuclei in the rostral medulla, e.g., GC, IRZ and PGCL, demonstrated dramatically decreasing levels with increasing PCA at death (age effect: p = 0.010, 0.007 and 0.030, respectively) (Fig 2, Table 5). The effect of age was also evident, although to a lesser degree in key target nuclei; i.e., those receiving 5-HT inputs within the mid-medulla, the HG (p = 0.002) and the CEN (p = 0.045), and the rostral medulla, the S5 (p = 0.026) and DAO (p = 0.018). These findings suggest that 5-HT$_{1A}$ activity decreased with increasing PCA in infants not dying of SIDS (considered a normative population), an effect that was especially strong within 5-HT source regions. They further suggest that analyses of age effects were needed to fully interpret the data, which we subsequently performed.

### 5-HT$_{1A}$ receptor binding in SIDS Vs. age-related controls (PostKCOD) in the first postnatal year

We measured 5-HT$_{1A}$ binding in five 5-HT source nuclei and eight target nuclei of all SIDS infants (premature and term combined; n = 14), comparing binding to PostKCOD control infants (n = 10). 5-HT$_{1A}$ receptor binding in the HG (upper airway patency during sleep and waking) in SIDS infants was increased by 51% compared to PostKCOD controls (p = 0.026; Table 6, Fig 3A). In testing for age vs. diagnosis interaction, we found significantly decreased 5-HT$_{1A}$ binding in both the NTS (age x diagnosis: p = 0.005) and MAO (age x diagnosis: p = 0.03) as PCA increased in SIDS infants, but not in PostKCOD controls (Fig 3A, an effect driven by prematurity (see below). 5-HT$_{1A}$ binding was ~30 fmol/mg and 60 fmol/mg in the NTS and MAO, respectively, in the youngest SIDS infants, while in the older SIDS infants, binding averaged around 10 fmol/mg (Fig 3A). Our data with significant interaction effects suggests that SIDS is associated with higher binding compared to PostKCOD controls, but only at younger ages and only in infants born premature. As mentioned, we repeated the analysis of 5-HT$_{1A}$ binding in SIDS vs. controls with PreKCOD infants included in the control group, given that their brain

**Table 5. Effect of postconceptional age on $^3$H-8-OH-DPAT binding to 5-HT$_{1A}$ receptors in pre- and postdischarge KCOD combined: Combined South Africa and Northern Plains cohort.**

| | N | Mean +/- SE (fmol/mg) | Age | |
| --- | --- | --- | --- | --- |
| | | | Beta | p-value |
| **Mid Medulla** | | | | |
| RO/RMg | 20 | 32.4 +/- 3.5 | −0.24 | 0.484 |
| HG | 21 | 8.9 +/- 0.9 | −0.23 | **0.002** |
| DMX | 21 | 10.1 +/- 0.9 | −0.14 | 0.109 |
| NTS | 21 | 13.8 +/- 1.3 | −0.17 | 0.184 |
| S5 | 21 | 22.4 +/- 2.1 | −0.07 | 0.720 |
| CEN | 21 | 19.2 +/- 1.5 | −0.27 | **0.045** |
| ARC | 20 | 4.3 +/- 0.5 | −0.003 | 0.945 |
| PIO | 21 | 3.6 +/- 0.9 | −0.003 | 0.957 |
| MAO | 21 | 22.7 +/- 2.7 | −0.24 | 0.352 |
| **Rostral Medulla** | | | | |
| RO/RMg | 21 | 51. 8 +/- 5.4 | −0.86 | *0.086* |
| GC | 21 | 26.4 +/-2.7 | −0.63 | **0.010** |
| PGCL | 21 | 27.4 +/- 3.0 | −0.60 | **0.028** |
| IRZ | 21 | 26.7 +/- 2.7 | −0.64 | **0.007** |
| S5 | 21 | 22.5 +/- 2.4 | −0.49 | **0.026** |
| ARC | 20 | 5.8 +/- 0.7 | −0.01 | 0.874 |
| PIO | 21 | 3.9 +/- 0.5 | −0.03 | 0.613 |
| DAO | 20 | 22.8 +/- 2.8 | −0.57 | **0.018** |
| **Rostral Pons** | | | | |
| DR | 13 | 52.0 +/- 6.7 | 0.66 | 0.292 |
| MR | 15 | 41.0 +/- 6.3 | 0.29 | 0.626 |
| LC | 13 | 16. 8 +/- 2.1 | −0.01 | 0.954 |
| PO | 15 | 22.3 +/- 3.3 | 0.17 | 0.574 |
| BP | 15 | 6.3 +/- 1.3 | −0.04 | 0.750 |

Legend. Significant p-values (<0.05) are bolded. Marginal p-values (<0.1) are in italics.
Abbreviations: PCA, postconceptional age; Ave, average; SIDS, sudden infant death syndrome; HG, hypoglossal nucleus; DMX, dorsal motor nucleus of the vagus; NTS, nucleus of the solitary tract; S5, spinal trigeminal nucleus; CEN, centralis; PIO, principal inferior olive; MAO, medial accessory olive; ARC, arcuate nucleus, RO, raphe obscurus; GC, gigantocellularis; PGCL, paragigantocellularis lateralis; IRZ, intermediate reticular zone; DAO, dorsal accessory olive; DR, dorsal raphe; MR, median raphe; LC, locus coeruleus; PO, nucleus pontis oralis; BP, basis pontis.

development is on a continuum with infants dying post-discharge (PostKCOD). With the addition of PreKCOD binding data, significance within the 3 nuclei remained (Fig 3B). This is the case for most of the nuclei we assessed in this study (see S1 Table 6). We analyzed 5-HT$_{1A}$ binding in the South African cohort alone and found no major discrepancies with results using the entire cohort (S1 Table 7).

### 5-HT$_{1A}$ receptor binding in the preterm cohort of the SPS

Among the SIDS infants in the SPS cohort, we recognized a subset of preterm infants. Given the effect of age on 5-HT$_{1A}$ binding in nuclei of infants not classified as SIDS, i.e., baseline controls, as well as the selective effect of PCA on 5-HT$_{1A}$ binding in the NTS and MAO of SIDS infants, we compared 5-HT$_{1A}$ binding in nuclei of premature SIDS cases against

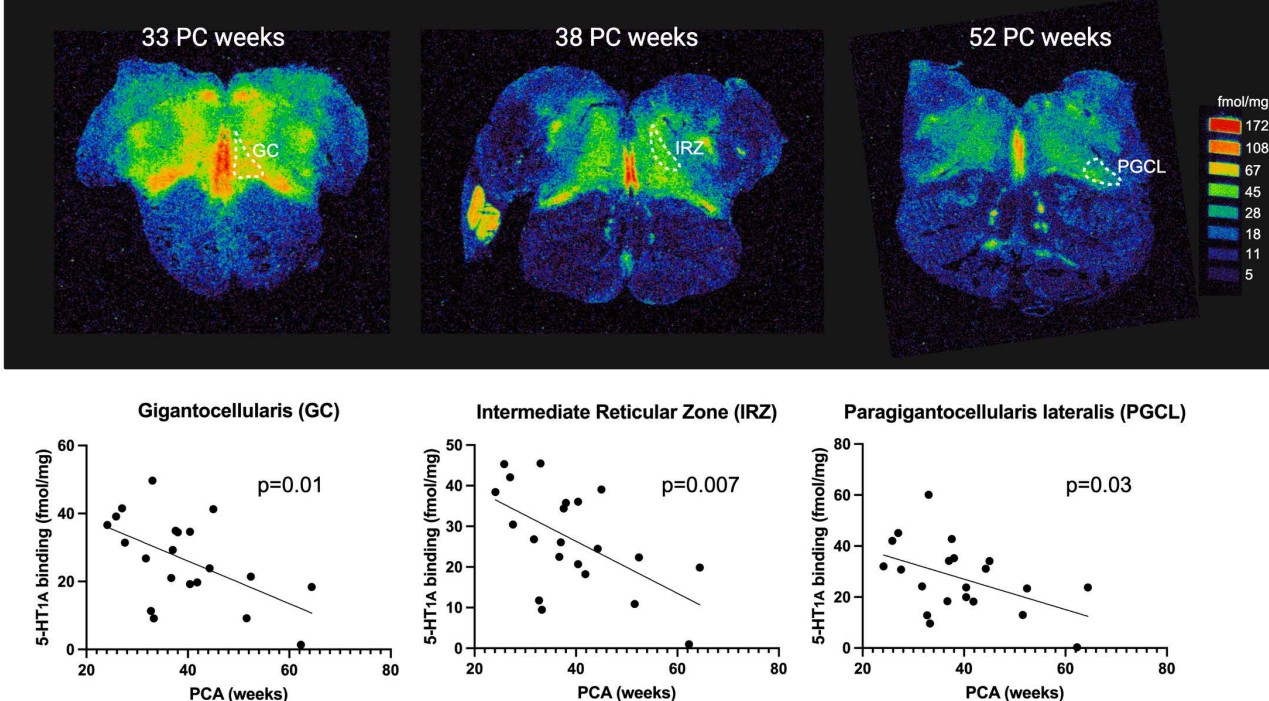

**Fig 2. Developmental change in 5-HT$_{1A}$ binding in known cause of death controls (predischarge and postdischarge combined).** Shown are autoradiographic images of 5-HT$_{1A}$ binding in the rostral medulla in three representative controls at 33, 38, and 52 post conceptional (PC) weeks. Three source nuclei that synthesize 5-HT are delineated with white lines (GC, gigantocellularis; IRZ, intermediate reticular zone; and PGCL, paragigantocellularis). Binding in these nuclei are visually (top row) and quantitatively (bottom row) decreasing with PC age, the latter shown with plots of 5-HT$_{1A}$ binding vs PC age. The relationship of PC age with binding is statistically significant in each of the nuclei shown. Data include the combined South Africa and Northern Plains cohorts.

premature controls and, separately, between term SIDS and term control infants. Compared to premature controls, 5-HT$_{1A}$ binding was higher in the NTS (p = 0.001), DMX (p = 0.038), HG (p = 0.007), MAO (p = 0.016), and CEN (p = 0.02) of SIDS infants born prematurely (Table 7, Fig 4). In contrast, compared to term controls, 5-HT$_{1A}$ binding was the same in these nuclei of SIDS infants born at term (Table 7, Fig 4).

## Effect of gestational exposures on maternal drinking and smoking on 5-HT$_{1A}$ binding

Data on maternal smoking and drinking during pregnancy are shown in Table 3 and S1 Tables 3–5. We examined whether there was a relationship between either the number of drinks per pregnancy on 5-HT$_{1A}$ binding or the average cigarettes per week during pregnancy on 5-HT$_{1A}$ binding. While there were no significant relationships seen in either analysis, our ability to adequately address the effect of exposure is limited due to the high rate of drinking and smoking in both SIDS and PostKCOD controls (Table 8) (See Limitations, strengths, and caveats).

## Discussion

SIDS is a major global health problem, with increased risk among socioeconomically disadvantaged populations who experience heightened stress. These populations include the American Indians in the Northern Plains, the Cape Coloured (mixed ancestry) in Cape Town, South Africa, African Americans in the United States, Maoris in New Zealand, and Aboriginal and Torres Strait Islanders in Australia [4–7]. Emerging autopsy data over the last two decades link at least a subset

**Table 6. Effect of diagnosis controlling for PCA on ³H-8-OH-DPAT binding in the brainstem: Combined South Africa and Northern Plains cohorts.**

| | Diagnosis | | | | | p-value | PCA | | Dx by PCA Interaction |
|---|---|---|---|---|---|---|---|---|---|
| | SIDS | | PostKCOD | | | | Beta | p-value | p-value |
| | N | Mean±SE | N | Mean±SE | | | | | |
| **Mid Medulla** | | | | | | | | | |
| RO/RMg | 12 | 30.9±4.5 | 9 | 31.4±5.2 | | 0.940 | −0.57 | *0.090* | 0.528 |
| HG | 13 | 10.4±0.9 | 10 | 6.9±1.1 | | **0.026** | −0.26 | **0.001** | 0.122 |
| DMX | 12 | 11.7±1.3 | 10 | 8.9±1.5 | | 0.182 | −0.18 | *0.071* | 0.222 |
| NTS | 13 | | 10 | | | | | | **0.005** |
| S5 | 13 | 27.2±3.1 | 10 | 23.6±3.5 | | 0.460 | −0.48 | **0.050** | 0.334 |
| CEN | 13 | 19.9±1.9 | 10 | 16.8±2.1 | | 0.283 | −0.61 | **<0.001** | 0.108 |
| ARC | 12 | 4.3±0.8 | 9 | 4.3±1.0 | | 0.980 | −0.07 | 0.250 | 0.382 |
| PIO | 13 | 4.0±0.8 | 10 | 3.4±0.9 | | 0.630 | −0.09 | 0.126 | *0.092* |
| MAO | 13 | | 10 | | | | | | **0.027** |
| **Rostral Medulla** | | | | | | | | | |
| RO/RMg | 14 | 41.0±5.27 | 10 | 42.8±6.1 | | 0.827 | −1.18 | **0.006** | 0.382 |
| GC | 14 | 23.3±2.7 | 10 | 21.5±3.2 | | 0.668 | −0.81 | **<0.001** | 0.838 |
| PGCL | 14 | 24.3±2.8 | 10 | 21.3±3.3 | | 0.504 | −0.80 | **0.001** | 0.374 |
| IRZ | 14 | 23.4±2.6 | 10 | 22.2±3.0 | | 0.780 | −0.83 | **<0.001** | 0.829 |
| S5 | 14 | 21.5±2.7 | 10 | 18.1±3.2 | | 0.418 | −0.69 | **0.002** | 0.423 |
| ARC | 14 | 6.3±0.9 | 9 | 5.6±1.1 | | 0.610 | −0.17 | **0.017** | 0.184 |
| PIO | 14 | 4.3±0.6 | 10 | 3.3±0.8 | | 0.359 | −0.10 | *0.051* | 0.102 |
| DAO | 14 | 18.8±2.3 | 10 | 19.2±2.7 | | 0.907 | −0.57 | **0.003** | 0.850 |
| **Rostral Pons** | | | | | | | | | |
| MR | 8 | 45.2±13.6 | 4 | 39.5±19.3 | | 0.812 | −0.06 | 0.951 | 0.488 |
| LC | 7 | 6.7±2.6 | 4 | 14.8±3.4 | | *0.090* | 0.09 | 0.615 | 0.107 |
| PO | 8 | 11.0±5.4 | 4 | 21.0±7.7 | | 0.305 | 0.03 | 0.943 | 0.173 |
| DR | 6 | 52.3±15.3 | 4 | 49.8±18.0 | | 0.914 | 0.79 | 0.432 | 0.468 |
| BP | 9 | 5.0±1.8 | 4 | 7.4±2.8 | | 0.470 | −0.15 | 0.291 | 0.818 |

Legend. Significant p-values (< 0.05) are bold. Marginal p-values (< 0.1) are in italics. Abbreviations: PCA, postconceptional age; Ave, average; SIDS, sudden infant death syndrome; HG, hypoglossal nucleus; DMX, dorsal motor nucleus of the vagus; NTS, nucleus of the solitary tract; S5, spinal trigeminal nucleus; CEN, centralis; PIO, principal inferior olive; MAO, medial accessory olive; ARC, arcuate nucleus, RO/RMg, raphe obscurus/Raphe magnus; GC, gigantocellularis; PGCL, paragigantocellularis lateralis; IRZ, intermediate reticular zone, DAO, dorsal accessory olive; DR, dorsal raphe; MR, median raphe; LC, locus coeruleus; PO, nucleus pontis oralis; BP, basis pontis.

of SIDS to abnormalities within the medullary serotonergic system, a system that contributes to cardiorespiratory and temperature homeostasis in sleep, promotes arousal, and supports critical respiratory and cardiovascular components of autoresuscitation, a life-preserving response to asphyxia and hypotension [10–20,75,76]. The analyses of the SPS demise cohort addressed two specific questions regarding the neuropathology involving 5-HT$_{1A}$ receptors in SIDS: 1) is reduced 5-HT$_{1A}$ binding apparent in infants dying of SIDS who lived in socioeconomic disadvantaged conditions, as has been found in San Diego cohorts of infants? and 2) does reduced 5-HT$_{1A}$ binding in SIDS associate with prenatal exposures to maternal drinking and smoking? We were unable to test the second hypothesis because smoking and drinking were as ubiquitous among the mothers of infants with known causes of death as those who died of SIDS. On the other hand, the high prevalence of premature birth in this population allowed a post-hoc exploration of the relationship between prematurity and SIDS, which was not included in the *a priori* analysis plan.

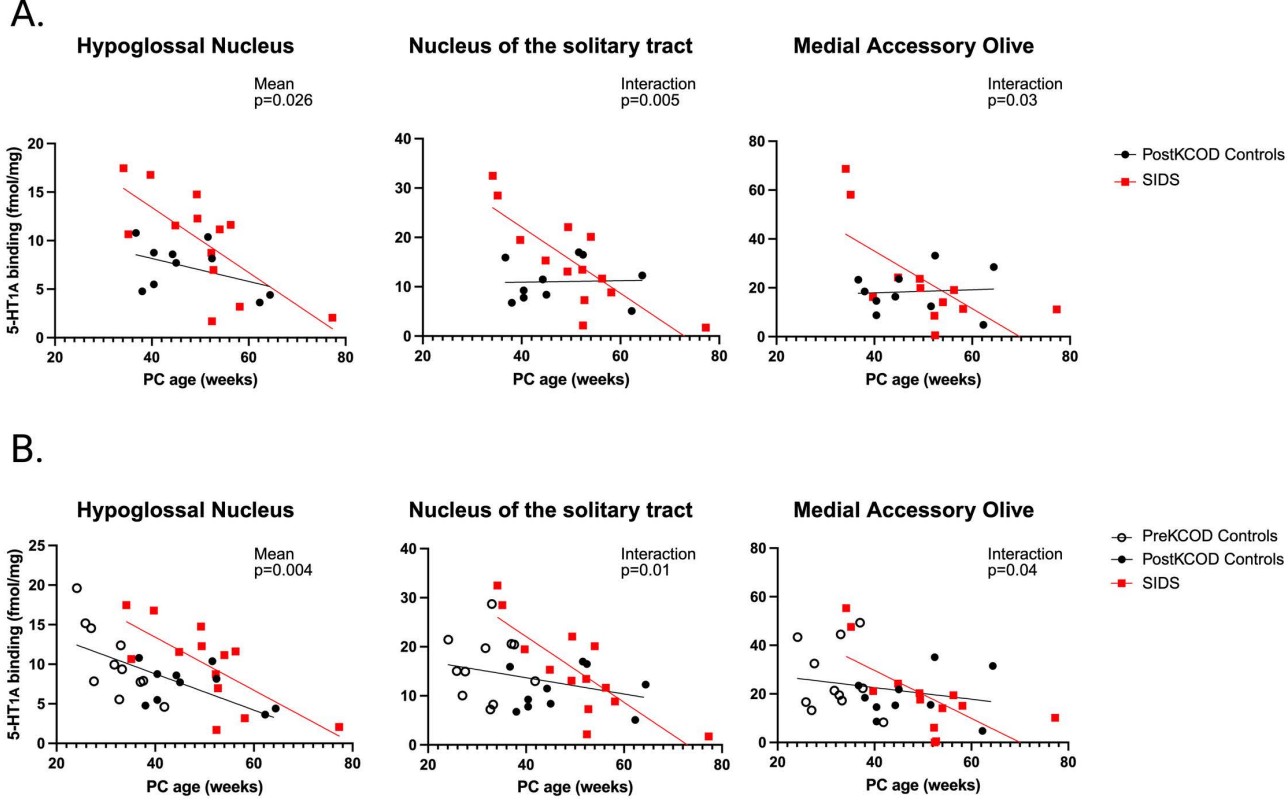

**Fig 3. 5.HT$_{1A}$ binding difference between known cause of death (KCOD) controls and SIDS.** Three nuclei measured at the mid-medulla level show abnormalities in SIDS compared to PostKCOD controls (A) and Pre- and PostKCOD controls combined (B). A) When SIDS is compared to PostKCOD controls only, the hypoglossal nucleus (HG) shows a significant increase in binding in SIDS compared to controls while the nucleus of the solitary tract (NTS) and medial accessory olive (MAO) show a significant age-vs-diagnosis interaction. In the NTS and MAO, note the decrease in binding with post-conceptional (PC) age in SIDS with no change in binding with age in the controls. B). When SIDS is compared to all KCOD controls, the HG, NTS, and MAO show a significant mean difference in 5-HT$_{1A}$ receptor binding. Data include the combined South Africa and Northern Plains cohorts.

As in our previous cohorts in San Diego [28,29], we report in the SPS cohort evidence of altered 5-HT$_{1A}$ binding in medullary nuclei critical for infant survival during episodes of asphyxia or hypotension. These new findings reinforce the concept that SIDS is a manifestation of one or more serotonopathies, irrespective of the specific population in which it occurs. Nevertheless, individual features of the serotonopathy were different between the SPS and San Diego cohorts, perhaps because the incidence of prematurity was higher in the SPS. While unexpected, these differences may provide important clues regarding the role of serotonergic defects in the pathogenesis of SIDS. In the following discussion, we focus on the roles of prematurity and maternal stress related to poor socioeconomic conditions on the development of the brainstem 5-HT system to garner fresh insight into possible mechanisms of sleep-related sudden death.

**Prematurity and 5-HT$_{1A}$ binding in the demise cohort of the SPS**

In contrast to previous studies, which generally show an age by diagnosis interaction (5-HT$_{1A}$ receptor binding declines as a function of PCA and is generally lower in infants who died of SIDS), 5-HT$_{1A}$ receptor binding in the SPS was significantly increased, not decreased, in the HG of SIDS infants compared to controls. This effect was largely driven by high binding in premature SIDS infants who died at a younger post-conceptional age (PCA), whereas SIDS infants born at term had the same 5-HT$_{1A}$ binding as controls. Compared to term SIDS infants, higher 5-HT$_{1A}$ receptor binding was also evident in

**Table 7. Analysis of ³H-8OH-DPAT binding in SIDS and PostKCOD controls born prematurely at term: Combined South Africa and Northern Plains cohorts.**

| | PRETERM | | | | | | | TERM | | | | | | |
|---|---|---|---|---|---|---|---|---|---|---|---|---|---|---|
| | n | Dx (SIDS vs PostKCOD) | | PCA | | Inter-action | | n | Dx (SIDS vs PostKCOD) | | PCA | | Inter-action |
| | | Beta | p-value | Beta | p-value | | | | Beta | p-value | Beta | p-value | |
| **Mid Medulla** | | | | | | | | | | | | | |
| RO/RMg | 11 | 3.23 | 0.766 | −0.22 | 0.792 | 0.149 | | 10 | −5.94 | 0.545 | −0.17 | 0.742 | 0.822 |
| HG | 11 | 5.94 | **0.007** | −0.11 | 0.422 | 0.841 | | 12 | 0.74 | 0.724 | −0.21 | *0.094* | 0.519 |
| DMX | 11 | 5.52 | **0.038** | 0.20 | 0.281 | 0.884 | | 11 | −0.99 | 0.703 | −0.15 | 0.309 | 0.412 |
| NTS | 11 | 11.70 | **0.001** | −0.84 | **0.002** | 0.308 | | 12 | −0.06 | 0.988 | −0.32 | 0.192 | 0.334 |
| S5 | 11 | 6.72 | 0.371 | 0.29 | 0.622 | 0.277 | | 12 | −1.31 | 0.795 | −0.17 | 0.549 | 0.456 |
| CEN | 11 | 8.20 | **0.021** | −0.62 | **0.024** | 0.387 | | 12 | −2.17 | 0.604 | −0.28 | 0.254 | 0.655 |
| ARC | 10 | 1.95 | 0.314 | −0.04 | 0.783 | 0.432 | | 10 | −5.94 | 0.545 | −0.17 | 0.742 | 0.822 |
| PIO | 11 | 3.16 | *0.054* | −0.08 | 0.480 | 0.650 | | 12 | −1.88 | 0.293 | −0.09 | 0.379 | 0.642 |
| MAO | 11 | 15.74 | **0.016** | −1.33 | **0.012** | 0.367 | | 12 | −10.47 | 0.130 | 0.18 | 0.622 | 0.680 |
| **Rostral Medulla** | | | | | | | | | | | | | |
| RO/RMg | 12 | 5.32 | 0.632 | −1.05 | 0.253 | *0.081* | | 12 | −12.01 | 0.261 | −0.09 | 0.883 | 0.727 |
| GC | 12 | 2.18 | 0.705 | −0.47 | 0.318 | 0.285 | | 12 | −0.51 | 0.922 | −0.28 | 0.344 | 0.857 |
| PGCL | 12 | 6.97 | 0.264 | −0.70 | 0.171 | 0.180 | | 12 | −2.76 | 0.619 | −0.21 | 0.503 | 0.841 |
| IRZ | 12 | 1.59 | 0.771 | −0.64 | 0.168 | 0.325 | | 12 | −0.99 | 0.854 | −0.32 | 0.314 | 0.848 |
| S5 | 12 | 5.65 | 0.392 | −0.62 | 0.252 | 0.539 | | 12 | 0.01 | 0.998 | −0.28 | 0.390 | 0.694 |
| ARC | 11 | 1.78 | 0.475 | −0.01 | 0.949 | 0.111 | | 12 | −0.82 | 0.526 | −0.07 | 0.374 | 0.380 |
| PIO | 12 | 2.27 | 0.200 | −0.10 | 0.468 | 0.410 | | 12 | −0.52 | 0.684 | −0.04 | 0.571 | 0.455 |
| DAO | 12 | −2.04 | 0.632 | −0.54 | 0.134 | 0.406 | | 12 | 0.04 | 0.994 | −0.15 | 0.655 | 0.702 |
| **Rostral Pons*** | | | | | | | | | | | | | |
| MR | 4 | | | | | | | 8 | 3.30 | 0.920 | 0.66 | 0.674 | |
| LC | 3 | | | | | | | 8 | −10.45 | *0.056* | 0.37 | 0.119 | |
| PO | 4 | | | | | | | 8 | −13.19 | 0.361 | 0.45 | 0.500 | |
| DR | 2 | | | | | | | 8 | −6.26 | 0.826 | 1.69 | 0.238 | |
| BP | 5 | | | | | | | 8 | −3.23 | *0.072* | 0.07 | 0.338 | |

**Legend.** Significant p-values (<0.05) are bold. Marginal p-values (<0.1) are in italics. No modelling was performed for bindings in the Rostral Pons of preterm infants due to insufficient sample size. Abbreviations: PCA, postconceptional age; Ave, average; SIDS, sudden infant death syndrome; HG, hypoglossal nucleus; DMX, dorsal motor nucleus of the vagus; NTS, nucleus of the solitary tract; S5, spinal trigeminal nucleus; CEN, centralis; PIO, principal inferior olive; MAO, medial accessory olive; ARC, arcuate nucleus, RO/RMg, raphe obscurus/Raphe magnus; GC, gigantocellularis; PGCL, paragigantocellularis lateralis; IRZ, intermediate reticular zone, DAO, dorsal accessory olive; DR, dorsal raphe; MR, median raphe; LC, locus coeruleus; PO, nucleus pontis oralis; BP, basis pontis. *Pontine data are presented for completeness, but we lack sufficient data from this region to draw conclusions at this time.

the NTS, MAO, DMX and CEN of premature SIDS infants. Despite differences between these data and those obtained from San Diego cohorts, it is notable that, irrespective of the cohort studied, altered 5-HT$_{1A}$ binding consistently appeared in the HG, DMX, NTS and MAO of SIDS infants and tended to diminish as the PCA at death increased for SIDS infants (see Fig 3).

What could be driving the differences that we observe in the current cohort, compared to those from San Diego? The distributions of race, ethnicity, and socioeconomic status are distinct in the SPS cohort. Notably, the rate of prematurity – defined as a birth that occurs before the 37th completed post-conceptional week – was especially high in control and SIDS SPS cohorts (~50%) compared to both the San Diego cohorts (~20%) as well as the rate of prematurity among

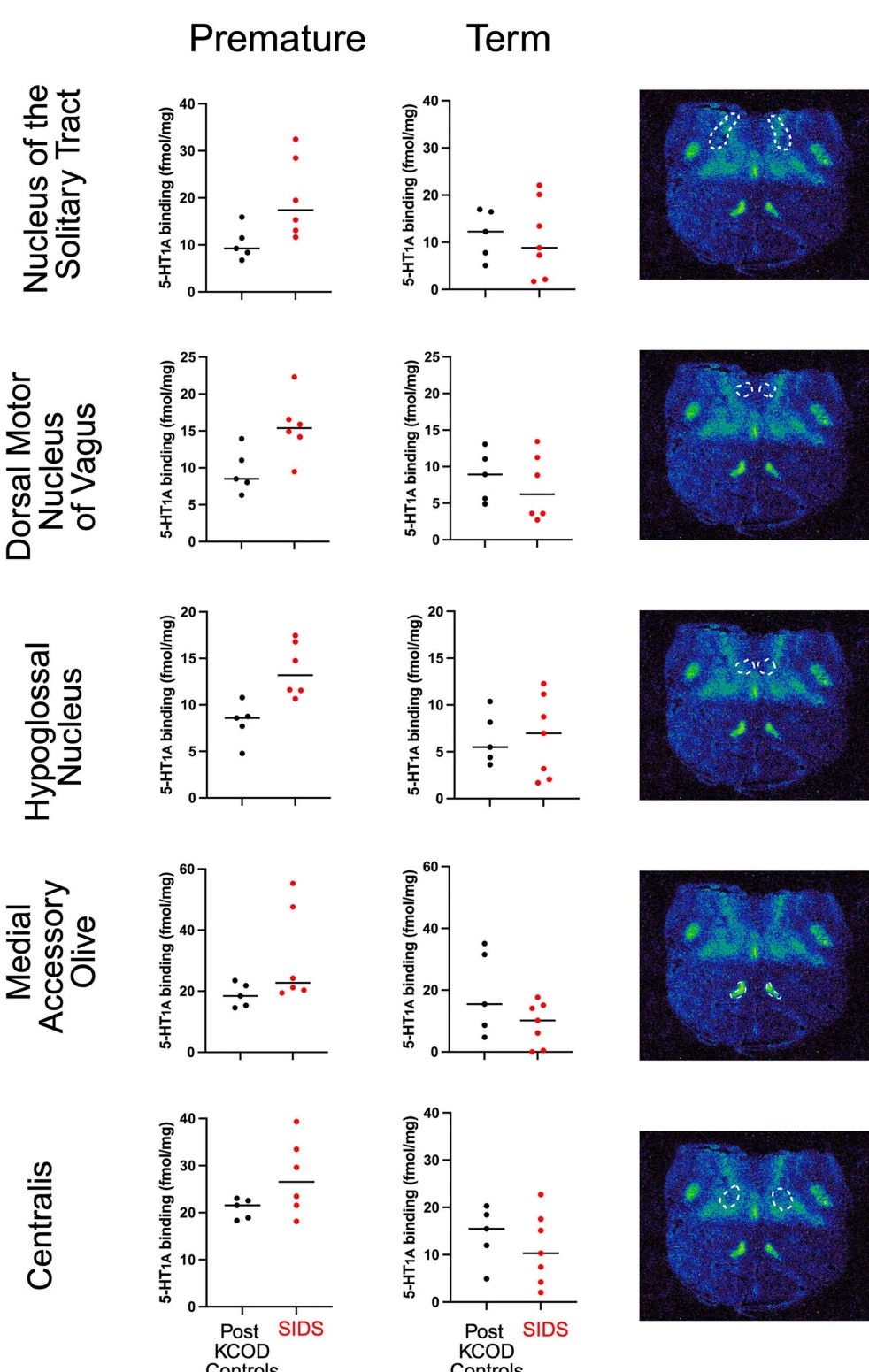

**Fig 4. Significantly different 5-HT$_{1A}$ receptor binding based on prematurity.** Premature SIDS infants show higher binding compared to premature post-discharge known cause of death (PostKCOD) controls in the nucleus of the solitary tract, the dorsal motor nucleus of vagus, the hypoglossal nucleus, the medial accessory olive, and the centralis [Left column]. There is no difference between term SIDS and term PostKCOD controls in these

same nuclei [Middle column]. The localization of each nucleus is indicated by a white boundary line in a representative autoradiograph of a 64 postconceptional week PostKCOD control born at term birth [Right column]. Data include the combined South Africa and Northern Plains cohorts..

**Table 8. Effects of exposure on $^3$H-8-OH-DPAT binding in the brainstem of SIDS and PostKCOD controls combined: Combined South Africa and Northern Plains cohorts.**

| | Alcohol on $^3$H-8-OH-DPAT binding | | | | | Smoking on $^3$H-8-OH-DPAT binding | | | | |
|---|---|---|---|---|---|---|---|---|---|---|
| | n | Drinks per pregnancy | | PCA | | n | Ave. cigarettes per week | | PCA | |
| | | Beta | p-value | Beta | p-value | | Beta | p-value | Beta | p-value |
| **Mid Medulla** | | | | | | | | | | |
| RO/RMg | 21 | −0.07 | 0.339 | −0.64 | *0.057* | 18 | −0.07 | 0.698 | −0.62 | *0.078* |
| HG | 23 | 0.02 | 0.368 | −0.22 | **0.010** | 20 | 0.04 | 0.396 | −0.2 | **0.031** |
| DMX | 22 | 0.01 | 0.754 | −0.16 | 0.135 | 19 | −0.01 | 0.8049 | −0.15 | 0.214 |
| NTS | 23 | −0.01 | 0.815 | −0.39 | **0.012** | 20 | −0.03 | 0.687 | −0.3 | 0.214 |
| S5 | 23 | 0.03 | 0.553 | −0.42 | *0.083* | 20 | 0.09 | 0.460 | −0.52 | *0.057* |
| Cent | 23 | −0.02 | 0.568 | −0.61 | **<0.001** | 20 | −0.02 | 0.801 | −0.52 | **0.003** |
| ARC | 21 | 0.01 | 0.525 | −0.06 | 0.325 | 18 | −0.02 | 0.594 | −0.06 | 0.405 |
| PIO | 23 | 0.01 | 0.450 | −0.08 | 0.188 | 20 | −0.02 | 0.478 | −0.08 | 0.211 |
| MAO | 23 | −0.03 | 0.636 | −0.59 | **0.035** | 20 | −0.06 | 0.614 | −0.37 | 0.138 |
| **Rostral Medulla** | | | | | | | | | | |
| RO/RMg | 24 | −0.04 | 0.666 | −1.23 | **0.005** | 21 | 0.29 | 0.132 | −1.21 | **0.004** |
| GC | 24 | −0.05 | 0.280 | −0.85 | **<0.001** | 21 | 0.06 | 0.564 | −0.79 | **0.002** |
| PGCL | 24 | −0.05 | 0.301 | −0.83 | **<0.001** | 21 | 0.04 | 0.696 | −0.7 | **0.003** |
| IRZ | 24 | −0.06 | 0.204 | −0.87 | **<0.001** | 21 | 0.04 | 0.678 | −0.79 | **0.001** |
| S5 | 24 | −0.06 | 0.155 | −0.73 | **0.001** | 21 | −0.03 | 0.804 | −0.68 | **0.006** |
| ARC | 23 | 0.01 | 0.660 | −0.16 | **0.026** | 20 | 0.05 | 0.159 | −0.19 | **0.012** |
| PIO | 24 | 0.004 | 0.711 | −0.09 | *0.082* | 21 | 0.02 | 0.562 | −0.10 | *0.064* |
| DAO | 24 | −0.04 | 0.254 | −0.61 | **0.002** | 21 | 0.07 | 0.429 | −0.55 | **0.007** |
| **Rostral Pons** | | | | | | | | | | |
| MR | | −0.25 | 0.791 | −0.02 | 0.986 | | −0.24 | 0.650 | −0.03 | 0.972 |
| LC | | 0.06 | 0.762 | 0.08 | 0.698 | | −0.05 | 0.673 | 0.10 | 0.616 |
| PO | | −0.11 | 0.783 | 0.08 | 0.847 | | −0.23 | 0.284 | 0.09 | 0.804 |
| DR | | 1.35 | 0.20 | 0.06 | 0.959 | | 0.16 | 0.760 | 0.75 | 0.455 |
| BP | | 0.003 | 0.980 | −0.14 | 0.324 | | 0.04 | 0.631 | −0.15 | 0.297 |

Legend. Significant p-values (<0.05) are bold. Marginal p-values (<0.1) are in italics. Abbreviations: PCA, postconceptional age; Ave, average; SIDS, sudden infant death syndrome; HG, hypoglossal nucleus; DMX, dorsal motor nucleus of the vagus; NTS, nucleus of the solitary tract; S5, spinal trigeminal nucleus; CEN, centralis; PIO, principal inferior olive; MAO, medial accessory olive; ARC, arcuate nucleus, RO/RMg, raphe obscurus/Raphe magnus; GC, gigantocellularis; PGCL, paragigantocellularis lateralis; IRZ, intermediate reticular zone, DAO, dorsal accessory olive; DR, dorsal raphe; MR, median raphe; DR, dorsal raphe, LC, locus coeruleus; PO, nucleus pontis oralis; BP, basis pontis.

all pregnancies worldwide (10%). The high rate of prematurity was not anticipated in the study design, but in retrospect, it is perhaps not surprising, given that 60% of all premature births worldwide occur in Africa and South Asia [72] where socioeconomically disadvantaged populations suffer from inadequate prenatal care and poor diet and nutrition. This is especially true at Tygerberg Hospital, the medical community from which the South African cohort was drawn. Prematurity is the leading cause of mortality at birth worldwide and is a consistently identified risk factor for SIDS globally [77–80]. Moreover, SIDS risk increases as gestational age decreases, suggesting some form of dose-dependency in whatever factors(s) are associated with prematurity that in turn increase the risk for SIDS.

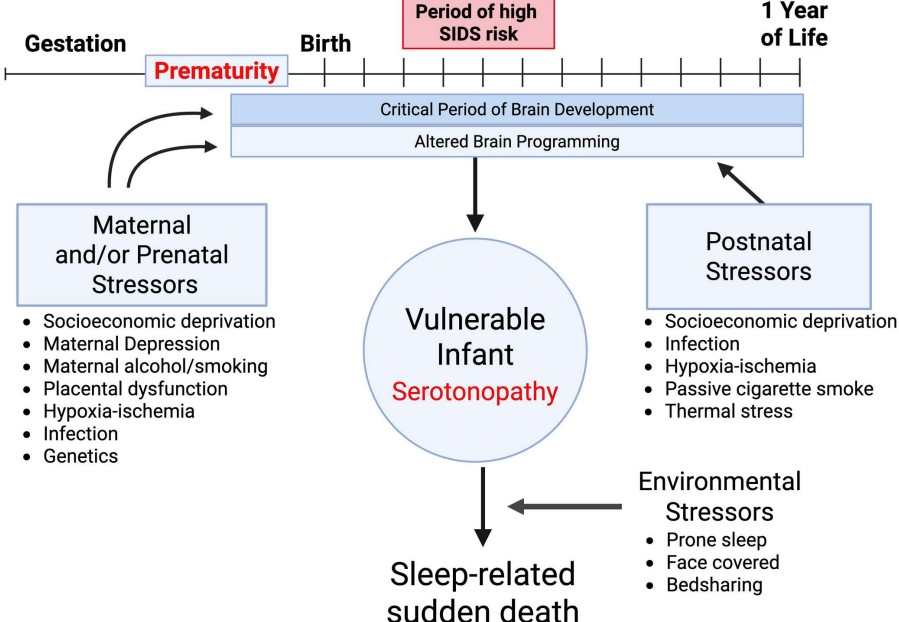

**Fig 5. Proposed expanded model of the pathogenesis of a medullary serotonopathy in a high-risk population of SIDS.** We propose that serotonergic receptor pathology originates among susceptible circuits involved in homeostatic control of autoresuscitation, sleep, and arousal during development of the brainstem (pons and medulla oblongata) in a subset of SIDS infants in a critical developmental period — a period that extends from the *prenatal* period into the *postnatal* period in the first year of life. This receptor pathology is a marker of an underlying brain vulnerability in the infant (serotonopathy) that compromises autoresuscitation during sleep upon confronting a life-threatening asphyxial challenge in the environment, as in unsafe sleep conditions. The expanded model that we are proposing is an elaboration of the original triple risk model for SIDS [105], which was based upon a multi-risk, multi-factorial premise of the pathogenesis of SIDS, involving the interlocking factors of the vulnerable infant, critical developmental period, and exogenous stressor in a three-circle Venn diagram. As shown in this figure, the revised model emphasizes the role of socioeconomic stressors for the mother and infant, incorporates subtypes of SIDS, one of which derives from a serotonopathy, and emphasizes the importance of prenatal risk factors. These new aspects of the model are based on novel information obtained mainly in the SPS, that highlights the vulnerability of high risk, socioeconomically depressed populations, which represent a large proportion of infants worldwide affected by SIDS. The revised model accounts for the high likelihood that the clinicopathologic phenotype of SIDS is due to multiple causes that form subsets of heterogeneous diseases, many yet to be discovered, in the vulnerable infant. In the revised model, medullary serotonopathies represent one such subtype. The revised model also emphasizes a fetal origin of the underlying vulnerability, with its lethal clinical manifestation after birth in the early postnatal period, recognized by the association of multiple prenatal stressors to SIDS in the SPS (see figure for list). A major concept introduced in the revised model of SIDS pathogenesis is the essential burden of stress and the social determinants of health in socially deprived populations transducing susceptible fetal circuits in gestation, disrupting fetal programming, and producing the underlying vulnerability. Postnatal stressors reinforce the burden of prenatal stressors (see list in figure). Thus, SIDS reflects an intricate constellation of internal and external factors, molded, in part, by stress, arising in fetal life and expressed in infancy. The importance of stress in the pathogenesis of SIDS represents a challenge and an opportunity that may require revamping political and scientific policy in SIDS research.

## Maternal stress related to socioeconomic conditions: impacts on the development of the 5-HT system and SIDS risk

A striking finding from this study was that infants who died of SIDS were born to mothers who may have experienced financial constraints and were more likely living in overcrowded, poor housing when compared to control mothers (Table 4). The effect of housing was most apparent in mothers who gave birth prematurely to infants who later died of SIDS. Sub-standard housing coupled with overcrowding likely leads to stress and possibly depression. Broadly speaking, "stress" is elicited when the mother or fetus is under homeostatic threat, leading to the activation of the hypothalamic-pituitary-adrenal (HPA) axis and, ultimately, elevated cortisol [81–84]. The response to acute stress is generally beneficial, eliciting a spectrum of biological adaptations aimed at threat mitigation [85]. Chronic or extreme stress, on the other

hand, results in dysregulation of the HPA axis and associated stress responses, leading to adverse effects including abnormal fetal brain development as well as premature birth [86,87]. Chronic stress and associated downstream neurophysiological responses – including those involving interactions between 5-HT and cortisol – can deflect the normal developmental trajectory of neural networks, including those containing serotonergic neurons [88–90]. Consequences may include altered proliferation and differentiation of 5-HT neurons, abnormal expression of 5-HT receptors, and aberrant connectivity and neurotransmission between serotonergic terminals and target neurons in key cardiorespiratory centers [88,89,91–99]. We are not able to analyze 5-HT$_{1A}$ binding in terms of indices of stress in this dataset because of the small numbers of cases. Nonetheless, we speculate that increased 5-HT$_{1A}$ binding may reflect increased receptor numbers, which could be related to impaired dendritic pruning and increased numbers of dendritic spines – all possible responses to stress. While not fully resolved, the mechanisms behind these developmental effects are likely multifactorial, involving elevated levels of cortisol and other stress hormones, intricate fetal-placental-maternal interactions, reprogramming of vulnerable fetal neural networks, and possibly epigenetic effects. Indeed, sub-optimal living conditions in animal studies induced epigenetic modification of 5-HT receptor expression [100–102]. We should also consider the potentially negative effects of maternal stress on fetal-placental function and consequently the fetal brain 5-HT system. According to many studies, the placenta provides 5-HT to the fetal CNS [103], and placental 5-HT contributes to proper fetal brain development and programming [86].

The concept of a fetal origin for SIDS fits with the Barker hypothesis and programming of the fetal brain, previously applied to other diseases of childhood and adulthood [104]. An altered developmental trajectory of the 5-HT system acquired *in utero* – i.e., increased 5-HT$_{1A}$ receptor activity for which we provide evidence here – may interact with known SIDS risk factors (e.g., cigarette smoke, infection, hypoxia) to further increase the risk of sudden death during a critical period of postnatal life. This is an extension of the Triple Risk Model [105], as shown here in Fig 5. The concept that stress *in utero*, whether originating maternally, from the placenta, or within the fetus itself, can alter the development of neurons in infancy has historical support; for example, SIDS is associated with immature development of dendritic spines [91,92] and delayed CNS myelination [106,107]. We postulate that stress represents a final common pathway of many SIDS risk factors including those associated with overcrowded housing and other factors related to social deprivation – e.g., maternal anxiety, smoking and drinking during pregnancy. Sudden death of an infant in a sleep period may occur when maternal stress coincides with environmental risk factors (e.g., prone sleep or bedsharing) and altered 5-HT$_{1A}$ activity or other intrinsic vulnerabilities.

## Social determinants of health in the SPS

The social determinants of health (SDOH) are the economic and social conditions that influence individual and group differences in health status [108]. They are the health promoting factors found in living and working conditions (such as the distribution of income, wealth, influence, and power), rather than individual risk factors (such as behavioral factors or genetics) that influence the risk or vulnerability for a disease. The distribution of social determinants is often molded by public policies that reflect the prevailing political ideologies of the area [109]. The World Health Organization stated that "the social determinants can be more important than health care or lifestyle choices in influencing health" and that "this unequal distribution of health-damaging experiences is not in any sense a 'natural' phenomenon but is the result of a toxic combination of poor social policies and economic arrangements" [110]. While their relative importance is controversial, major social determinants include gender, race, economics, education, sanitation, employment, housing, phone ownership, and food security [111–113]. In short, social determinants are non-medical factors that influence health outcome and have a direct correlation with health equity [108–110,114]. The association of chronic prenatal stress, which may be experienced more frequently by those living in adverse social and economic conditions, with SIDS may support our argument that prenatal stress leads to altered brain programming and poor health outcomes, some of which can be hypothesized to have an epigenetic basis. Thus, SIDS in the socially deprived, like the encephalopathy of prematurity, represents a

'complex amalgam' of biological, developmental, and environmental factors [115]. The eradication of the intricate, intertwined conglomeration of factors leading to SIDS will, therefore, require new knowledge emerging from rigorous, hypothesis-driven science as well as social policy reform to reduce the burden of poor socioeconomic conditions and its effects on maternal and infant health worldwide.

## Potential consequences in infant physiology of a medullary serotonopathy in the SPS

Several studies using neonatal rodents strongly suggest that the $5\text{-HT}_{1A}$ receptor is important for autonomic and respiratory function in early life [34,116]. $5\text{-HT}_{1A}$ is an inhibitory receptor expressed on 5-HT neuronal soma and dendrites (i.e., an autoreceptor) reducing their activity. It is also expressed on non-5-HT neurons (heteroreceptor), including inhibitory GABAergic and glycinergic neurons, where activation of $5\text{-HT}_{1A}$ can reduce inhibitory signaling. As with all 5-HT receptors, the activity of $5\text{-HT}_{1A}$ must be developmentally appropriate; increased activity – as reflected here with increased binding in SIDS infants born prematurely – can sway the balance of excitatory-inhibitory neurotransmission towards inhibition.

As we have previously reported in San Diego cohorts, we demonstrated abnormal $5\text{-HT}_{1A}$ binding in the HG, DMX, NTS and MAO, all of which are innervated by the medullary 5-HT system and have critical roles within integrated circuits for cardiorespiratory and temperature homeostasis, arousal and autoresuscitation in response to asphyxia and hypotension [32,117]. The HG serves a critical role in the stabilization and patency of the upper airway; specifically, it provides excitatory drive to the genioglossus muscle which facilitates the movement of the tongue during inspiration to reduce resistance in the upper airway. As an inhibitory receptor, increased $5\text{-HT}_{1A}$ activity in early postnatal life, as reflected by increased receptor binding, may lead to inhibition of the HG [118,119], reducing genioglossus activity and increasing upper airway resistance during sleep. Enhanced $5\text{-HT}_{1A}$ drive to cardiac vagal neurons in the DMX may alter parasympathetic-sympathetic balance to the heart and potentially increase the magnitude of potentially dangerous, reflex bradycardias in sleep [120]. The NTS and MAO are critical for arterial blood pressure homeostasis. The NTS receives a variety of visceral afferent inputs, including those originating from the arterial chemoreceptors and baroreceptors. 5-HT neurons heavily innervate the NTS and it has been well-established that 5-HT acting in the NTS can alter baroreflex function [121,122]. Infant animals lacking 5-HT have reduced blood pressure and a more severe drop in blood pressure during asphyxial conditions, owing to compromised sympathetic and parasympathetic nervous system activity [123]. Serotonergic neurons also abundantly innervate the MAO [124,125], a nucleus that facilitates the restoration of blood pressure in a variety of physiological and pathophysiological contexts, including those associated with changing body position (i.e., supine vs. prone) [126]. Altered $5\text{-HT}_{1A}$ activity in the NTS and MAO may therefore put an infant at risk for hypotensive events associated with the prone sleep position. We speculate that SIDS represents the outcome of several unique serotonopathies, each with its own neuropathological "signature" that reflects dysfunction within a particular combination of serotonergic source and/or target nuclei, yet share the common phenotype of sudden death in a sleep period during a window of postnatal life and, critically, when specific risk factors exist in the infant's environment.

## Limitations, strengths, and caveats

This analysis had several limitations, primarily concerning the small sample size, which is not uncommon in neuropathological studies in SIDS, due to logistical difficulties in accrual of autopsy brain tissue without lengthy postmortem intervals in SIDS and control infants. Due to the small sample sizes in this study, complex models could not be built to assess interactions of exposure and diagnosis on $5\text{-HT}_{1A}$ binding. Additional stratified analyses by preterm vs. term also could not be done. One of our original hypotheses was regarding the effect of prenatal exposures on the serotonergic system, specifically, $5\text{-HT}_{1A}$ binding in SIDS. However, a second limitation to the study was the fact that all mothers (SIDS and controls) smoked during pregnancy, and many drank alcohol. Statistically we saw no effect of drinks per pregnancy or number of cigarettes per week on $5\text{-HT}_{1A}$ receptor binding. Although this lack of effect may suggest that alcohol and smoking have limited effects on binding, it may also reflect an inability to meaningfully parse out any relationship between exposure and

binding, due to the near-ubiquitous use of cigarettes and alcohol. Our inability to ascertain differences between SIDS and known causes of death based on exposure differs from the epidemiological study of exposure and outcomes published in 2020 [23]. The previous study showed increased relative risks for SIDS compared to KCOD controls with combined smoking and drinking alone (relative risk of 3.95 and 4.86, respectively) and drinking and smoking in combination (relative risk of 11.79). Differences between the two studies include; (1) a smaller number of SIDS and KCOD controls in the current study relative to the epidemiological study of 2020 (14 vs. 28 SIDS and 10 vs. 38 PostKCOD controls); and (2) different methods used to characterize exposure during pregnancy (binary values of exposure [yes/no] and continuous variable of exposure per pregnancy and week vs. group-based trajectory modeling).

Also limiting to the current study is the fact that the control group consisted of infants who died of heterogeneous clinically recognized illnesses early in life, notably with systemic infection severe enough to be considered the cause of death by the pathology review panel (see above). These infants can be considered non-SIDS controls, but they should not be considered "normal" or necessarily "healthy" controls. Apart from accidental deaths, e.g., motor accidents, few "normal" infants die. Beyond the presence of serious illnesses in the control infants, virtually all the infants were also exposed to cigarette smoke and alcohol *in utero* in the SPS. Thus, non-SIDS cases are necessarily the default, but they are not an ideal control group.

Strengths of the study include high significance of the clinical problem worldwide, prospective data collection, rigorous adjudication of cause of death blinded to brainstem biochemical data by an experienced group of clinical and pathology investigators, brainstem biochemistry in properly prepared (non-fixed) tissues, multi-disciplinary collection and analysis of multiple clinical and demographic variables and invested research participants. An unanticipated strength was the accrual of a sufficient sample size of preterm SIDS and controls for reliable statistical comparisons—an almost unheard-of event in SIDS neuropathological studies based on single medical examiners' offices—but essential for giving insight into the role of preterm birth in the pathogenesis of altered 5-HT development in SIDS as reported in the demise SPS.

While not a weakness or limitation, our proposed conclusions come with the caveat that data were based on the single methodology of ligand receptor binding. Additional techniques are required to validate abnormalities of the serotonergic system in SIDS, including distinct abnormalities in SIDS infants related to prematurity. These techniques include specific examination of the HTR1A receptor gene and/or 5-HT$_{1A}$ protein and unbiased expression of dysregulated 5-HT-related genes via unbiased transcriptomics. Given the potential role of epigenetic modifications to 5-HT related genes, including HTR1A [100–102], an examination of these modifications is warranted.

## Conclusions and future directions

Despite worldwide public health campaigns encouraging safe (supine) infant sleep, SIDS remains a major cause of infant mortality, especially in socioeconomically deprived populations. Despite the somewhat disparate findings between the SPS cohort and those from San Diego, the findings in the SPS cohort lend further support to the concept that a least a subset of SIDS represents one or more serotonopathies [18,32]. In addition to high 5-HT$_{1A}$ binding in SIDS infants, a unique, unexpected feature of the current study was the high proportion of infants born prematurely, which drove our main findings, and may have uncovered a unique, and not previously well appreciated, subset or pathophysiological endotype of SIDS in premature infants with serotonin abnormalities distinct from SIDS infants born at term. We show that SIDS in our database is associated with poor housing, overcrowding and poverty, conditions potentially leading to maternal stress that may have set in motion the cascade of events, including developmental serotonergic reprogramming, compromising critical neural networks and increasing the risk of sudden death during sleep within a vulnerable period of postnatal life. Although the concept requires experimentation in animals, we speculate that maternal, prenatal, and postnatal stress alter the trajectory of brain development and modify the expression of 5-HT$_{1A}$ and other serotonergic receptors, increasing the risk of sudden death. It seems likely that as the burden of stress increases, brain development progressively deviates from the normal trajectory, increasing the risk of SIDS. While the results of the study are readily generalizable to high-risk

populations with prenatal smoking and alcohol intake and prematurity, the results suggest that pre- and/or postnatal stress is a common denominator, with the overall nature and burden of the stress being unique in each population.

The ubiquity of social stressors and poverty in the SPS mandates a rethinking of our models of the pathogenesis of SIDS. Stress is universal and exists in various forms, extending from the Cape Coloured, low-income populations in South Africa to low- and middle-income communities of San Diego County. Yet, social deprivation more often leads to additive, prolonged, and/or severe stress; its cumulative effects may account for the increase in the odds ratio for SIDS in socio-economically disadvantaged populations [4,5]. The multidisciplinary findings of the SPS suggest several broad areas for ongoing and future research in SIDS. Of paramount importance is the generation of new knowledge regarding the effects of maternal stress across multiple income brackets, on infant health and the risk for SIDS and other causes of infant mortality. We need clarification on how stress impacts the development and fetal programming of the medullary serotonergic and other neurotransmitter systems, focusing on all income brackets. Further research is also needed to resolve the mechanisms underlying the strong association of SIDS with prematurity and, more generally, whether SIDS is comprised of subsets of yet-to-be-identified diseases with heterogeneous etiologies.

As the SPS cohort is unique in its demographics and associated risk factors, it is perhaps not surprising that it displays a unique serotonergic signature. While the SPS cohort has limitations (i.e., small numbers, lack of healthy controls, etc), as does any autopsy study, an important point is that medullary serotonergic circuits are altered in all SIDS cohorts that we have studied to date, irrespective of global background. A better understanding of the basic pathological mechanisms, including how environmental, maternal, and *in utero* risk factors interact is required if we are to develop prophylactic strategies to eliminate SIDS globally. This will undoubtedly require sweeping social reform, universal access to health care, the development of potential pharmacological interventions and biomarkers in living infants, and promotion of current and future risk reduction strategies.

## Supporting information

**S1 File. Supplemental Tables.** Supplemental Tables 1–5 provide additional information regarding alcohol and smoking exposure in different cohorts examined in the study. Supplemental Table 6 provides $^3$H-8-OH-DPAT binding data in SIDS vs all KCOD controls (pre- and postdischarge). Supplemental Table 7 provides $^3$H-8-OH-DPAT binding data in the South African cohort only. Supplemental Table 8 provides cohort information across tables and figures.
(PDF)

**S2 File. Kinney et al 5-HT1A ligand binding data.** $^3$H-8-OH-DPAT binding data is provided for all medullary nuclei measured. Binding is listed in fmol/mg. Cases are identified with the DASH ID consistent with the DASH database (see Datasharing statement). Data has not been included for 3 cases from the American Indian tribes in the Nothern Plains. Please see the Datasharing Statement for information.
(XLSX)

## Acknowledgments

We thank the families and communities for their courageous commitment to the Safe Passage Study. We thank Kimberly Dukes, PhD, and Lisa Sullivan, PhD, for their work in the Safe Passage Study and statistical guidance. We thank Katie Ziegler for her commitment to the work of the Safe Passage Study and for assistance in database management. We remember with great fondness our beloved mentor and friend, the late Professor Richard H. Hewlett, Neuropathologist at Tygerberg Hospital and Stellenbosch University and Consultant at University of Cape Town; and the late Dr. Johan Dempers, Chief Forensic Pathologist for Western Cape, Consultant at Tygerberg Hospital and Stellenbosch University. The wisdom and guidance of these men were key in many points in the Safe Passage Study. We thank the late Elaine Geldenhuys for dedicated and outstanding technical assistance. From NIAAA, we remember the late Dr. Dale Hereld for

his scientific guidance and input and the late Dr. Kenneth Warren who was instrumental in supporting the Safe Passage Study.

## Author contributions

**Conceptualization:** Hannah C. Kinney, Amy Elliot, William Fifer, Hein Odendaal, Robin Haynes.

**Data curation:** Hannah C. Kinney, Rebecca Folkerth, Morgan Nelson, Lucy Brink, Robin Haynes.

**Formal analysis:** Morgan Nelson, Lucy Brink, Felicia Trachtenberg.

**Funding acquisition:** Hannah C. Kinney, Jyoti Angal, Amy Elliot, William Fifer, Howard Hoffman, Hein Odendaal, Robin Haynes.

**Investigation:** Hannah C. Kinney, Rebecca Folkerth, Jyoti Angal, Kevin Broadbelt, Theonia Boyd, Elsie Burger, Jean Coldrey, Kevin Cummings, Jhodie Duncan, Amy Elliot, William Fifer, Eugene Nattie, Laurie Nelsen, Hein Odendaal, David Paterson, Bradley Randall, Drucilla Roberts, Pawel Schubert, Mary Ann Sens, Shabbir Wadee, Colleen Wright, Dan Zaharie, Robin Haynes.

**Methodology:** Kevin Broadbelt, Jhodie Duncan, David Paterson, Robin Haynes.

**Project administration:** Hannah C. Kinney, Jyoti Angal, Amy Elliot, William Fifer, Hein Odendaal, Robin Haynes.

**Resources:** Hannah C. Kinney, Jyoti Angal, Amy Elliot, William Fifer, Hein Odendaal, Robin Haynes.

**Supervision:** Hannah C. Kinney, Jyoti Angal, Amy Elliot, William Fifer, Hein Odendaal, Robin Haynes.

**Validation:** Hannah C. Kinney, Morgan Nelson, Amy Elliot, William Fifer, Hein Odendaal, Robin Haynes.

**Visualization:** Hannah C. Kinney, Kevin Cummings, James Leiter, Robin Haynes.

**Writing – original draft:** Hannah C. Kinney, Felicia Trachtenberg, Kevin Cummings, James Leiter, Robin Haynes.

**Writing – review & editing:** Hannah C. Kinney, Rebecca Folkerth, Morgan Nelson, Kevin Cummings, Amy Elliot, Howard Hoffman, James Leiter, Eugene Nattie, Hein Odendaal, Robin Haynes.

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
