## [Decision Letter · Decision Letter 0]

2 Jun 2025

PONE-D-25-13478Serotonergic Receptor Binding in the Brainstem in the Sudden Infant Death Syndrome in a High-Risk PopulationPLOS ONE

Dear Dr. Haynes,

Thank you for submitting your manuscript to PLOS ONE. After careful consideration, we feel that it has merit but does not fully meet PLOS ONE’s publication criteria as it currently stands. Therefore, we invite you to submit a revised version of the manuscript that addresses the points raised during the review process.

**ACADEMIC EDITOR:** The study is well-designed and executed. I appreciate the authors for pointing out the observed strength and limitations of the study. However, the reviewers have identified some grey areas that require author's significant attention before the manuscript can be processed further. I hereby recommend a major revision to resolve the concerns. 

We look forward to receiving your revised manuscript.

Kind regards,

Yusuf Oloruntoyin Ayipo, Ph.D

Academic Editor

PLOS ONE

Journal Requirements:

[The research reported in this publication was supported by National Institutes of Health (NIH) grants U01HD045935, U01HD055155, U01HD045991, and U01AA016501 funded by the National Institute on Alcohol Abuse and Alcoholism (https://www.niaaa.nih.gov/), Eunice Kennedy Shriver National Institute of Child Health and Human Development (https://www.nichd.nih.gov/), and the National Institute on Deafness and Other Communication Disorders (https://www.nidcd.nih.gov/).  This was an international consortium and all authors were member of the consortium.].

3. Thank you for stating the following in your manuscript:

[The research reported in this publication was supported by National Institutes of Health (NIH) grants U01HD045935 (Elliot), U01HD055155 (Fifer), U01HD045991 (Kinney), and U01AA016501 (Odendaal) funded by the National Institute on Alcohol Abuse and Alcoholism, *Eunice Kennedy Shriver*  National Institute of Child Health and Human Development, and the National Institute on Deafness and Other Communication Disorders.]

[The research reported in this publication was supported by National Institutes of Health (NIH) grants U01HD045935, U01HD055155, U01HD045991, and U01AA016501 funded by the National Institute on Alcohol Abuse and Alcoholism (https://www.niaaa.nih.gov/), Eunice Kennedy Shriver National Institute of Child Health and Human Development (https://www.nichd.nih.gov/), and the National Institute on Deafness and Other Communication Disorders (https://www.nidcd.nih.gov/).  This was an international consortium and all authors were member of the consortium.]

5. Please amend the manuscript submission data (via Edit Submission) to include author Hannah C Kinney MD.

6. Please amend your authorship list in your manuscript file to include author Hannah L Kinney.

7. We note that you have included the phrase “data not shown” in your manuscript. Unfortunately, this does not meet our data sharing requirements. PLOS does not permit references to inaccessible data. We require that authors provide all relevant data within the paper, Supporting Information files, or in an acceptable, public repository. Please add a citation to support this phrase or upload the data that corresponds with these findings to a stable repository (such as Figshare or Dryad) and provide and URLs, DOIs, or accession numbers that may be used to access these data. Or, if the data are not a core part of the research being presented in your study, we ask that you remove the phrase that refers to these data.

Additional Editor Comments:

The study is well-designed and executed. I appreciate the authors for pointing out the observed strength and limitations of the study. However, the reviewers have identified some grey areas that require author's significant attention before the manuscript can be processed further. I hereby recommend a major revision to resolve the concerns.

Reviewers' comments:

Reviewer's Responses to Questions

**Comments to the Author**

1. Is the manuscript technically sound, and do the data support the conclusions?

Reviewer #1: Yes

Reviewer #2: Yes

Reviewer #3: Yes

Reviewer #4: Yes

2. Has the statistical analysis been performed appropriately and rigorously? 

Reviewer #1: Yes

Reviewer #2: Yes

Reviewer #3: Yes

Reviewer #4: Yes

3. Have the authors made all data underlying the findings in their manuscript fully available?

Reviewer #1: Yes

Reviewer #2: Yes

Reviewer #3: Yes

Reviewer #4: Yes

4. Is the manuscript presented in an intelligible fashion and written in standard English?

Reviewer #1: Yes

Reviewer #2: Yes

Reviewer #3: Yes

Reviewer #4: Yes

5. Review Comments to the Author

Reviewer #1: Thank you for this thorough and meaningful contribution to our understanding of the neuropathophysiology underlying SIDS in high-risk populations. Your study provides important evidence supporting the serotonopathy hypothesis in SIDS, building on prior findings with a novel cohort characterized by high rates of maternal substance exposure and socioeconomic disadvantage.

Strengths of the manuscript include:

Use of prospective, well-characterized cohort data from the Safe Passage Study.

Detailed receptor binding analysis across multiple brainstem regions.

Consideration of developmental trajectories and the effect of prematurity.

Thoughtful discussion contextualizing findings within public health and biological frameworks.

Areas for improvement:

Clarify the timing for the public release of supplemental receptor binding data.

Explicitly state in the main manuscript whether any outliers or missing data were excluded and how.

In the discussion, more explicitly address why some expected correlations (e.g., with maternal smoking or alcohol use) were not statistically significant, despite high exposure prevalence.

Reviewer #2: This manuscript presents a rigorous and thoughtful investigation into serotonergic receptor binding in the brainstems of infants who died of SIDS in a high-risk population. The authors make a meaningful contribution to the field by replicating prior findings of serotonergic abnormalities and extending them to socioeconomically disadvantaged populations that are often underrepresented in neurobiological studies.

The study is strengthened by its careful control of developmental age (PCA), the inclusion of both pre- and post-discharge control groups, and the integration of socioeconomic and environmental variables such as maternal substance use, housing conditions, and maternal mental health. The authors appropriately acknowledge the limitations of their sample size and the difficulty in disentangling the impact of prenatal exposures in populations with uniformly high smoking and alcohol exposure rates. Importantly, the authors avoid overstating these associations and interpret their findings with appropriate caution. Furthermore, the replication of serotonergic abnormalities in a global cohort, combined with data on socioeconomic and prenatal exposures, offers insights with broad implications for both basic science and health equity.

The data are clearly presented, the methodology is thorough, and the statistical approach is sound. The manuscript is well-organized and written in a clear and accessible manner. The findings support and refine the “serotonopathy” hypothesis of SIDS by demonstrating age-related changes in 5-HT1A binding and identifying brainstem with differential patterns in SIDS cases compared to controls. These insights have relevance for understanding the neurobiology of SIDS in diverse populations and could inform future public health strategies or mechanistic investigations.

I have no major concerns. This is a well-executed study.

Reviewer #3: The manuscript contributes valuable new insights into the neuropathology of SIDS, particularly in high-risk, socioeconomically vulnerable populations. The observed elevation of 5-HT1A receptor binding in premature SIDS cases challenges previous assumptions and suggests potential subtypes of serotonopathy. While the study is well-executed and rigorously presented, few clarifications, analyses, and interpretive expansions are needed before publication.

Reviewer #4: The manuscript addresses an important scientific question and is based on rich and valuable histopathological data. The statistical approach is fundamentally sound, particularly in its use of regression models and adjustments for postconceptional age; however, it requires stronger justification and more transparent reporting, especially in relation to multiple comparisons and the modeling of small subgroups. The interpretation of exposure-related findings should be more cautious, as the cohort’s near-universal exposure to smoking and alcohol limits the ability to draw meaningful conclusions. Additionally, some claims made in the discussion overstate the generalizability of the findings and should be revised for a more measured tone. With these revisions, the manuscript would be suitable for publication.

6. PLOS authors have the option to publish the peer review history of their article (what does this mean? ). If published, this will include your full peer review and any attached files.

**Do you want your identity to be public for this peer review?** For information about this choice, including consent withdrawal, please see our Privacy Policy .

Reviewer #1: No

Reviewer #2: **Yes: ** Margaret Ebun Olawale

Reviewer #3: No

Reviewer #4: No

---

## [Author Response · Author response to Decision Letter 1]

16 Jul 2025

Below we respond to each of the concerns/suggestions of the reviewers according to your instructions. These are also noted in the cover letter.

1. We have ensured that our manuscript meets PLOS One’s style requirement.

2. Funding Statement. We have removed the funding statement from the manuscript and placed it in this cover letter as instructed (see below).

Revised Funding Statement:

The research reported in this publication was supported by National Institutes of Health (NIH) grants U01HD045935 (MEN, JA, AJE, LLN, BBR, MAS ), U01HD055155 (WPF), U01HD045991 (HCK, RDF, KGB, JRD, DSP), and U01AA016501 (LB, EHB, JAC, HJO) funded by the National Institute on Alcohol Abuse and Alcoholism (https://www.niaaa.nih.gov/), Eunice Kennedy Shriver National Institute of Child Health and Human Development (https://www.nichd.nih.gov/), and the National Institute on Deafness and Other Communication Disorders (https://www.nidcd.nih.gov/). There was no additional external funding received for this study.

The work was performed in the context of the research network Prenatal Alcohol, SIDS, and Stillbirth (PASS) Network. This Network included affiliates of the NIH who participated in the overall PASS study design, decision to publish, and preparation of the manuscript.

3. Data Availability Statement

We have revised our data sharing statement (see below). Please note that we are unable to share data from tribal participants. There were 3 infants in our cohort from tribal communities in the Northern Plains.

Revised Statement: De-identified data from the Safe Passage Study is available through NICHD’s Data and Specimen Hub (DASH). All case demographic and exposure data are available on DASH. Elliott, Amy (2025). A Prospective Study on the Role of Prenatal Alcohol Exposure in SIDS and Stillbirth (Version 1). NICHD Data and Specimen Hub. https://doi.org/10.57982/sv8c-4y07. Ligand receptor binding data are available as Supplemental File 2. In compliance with tribal data sharing agreements, we have excluded 3 cases from American Indian tribes in the Northern Plains from these publicly shared datasets.

4. The instructions asked to amend the manuscript submission data (via Edit Submission) to include author Hannah C Kinney MD. We are unsure what is needed, since Hannah Kinney is listed as the first author under “Current Author List”

We have provided all of the data related to sleeping conditions in the text under “Clinicopathological Information”. We removed the “Data Not Shown” statement.

5. We have provided captions for our supporting information files at the end of the manuscript.

Reviewer #1.

• Clarify the timing for the public release of receptor binding data.

-The receptor binding data is now provided as a supplemental file (see the Data Availability Statement above)

• Explicitly state in the main manuscript whether any outliers or missing data were excluded and how.

-We have added the following statement to the statistical methods. “There were no outliers excluded from statistical analysis and complete case analysis was performed.”

• More explicitly address why some expected correlations (e.g., with maternal smoking or alcohol use) were not statistically significant, despite high exposure prevalence.

-We have added the following statement to the limitations section.

“One of our original hypotheses was regarding the effect of prenatal exposures on the serotonergic system, specifically, 5-HT1A binding in SIDS. However, a second limitation to the study was the fact that all mothers (SIDS and controls) smoked during pregnancy, and many drank alcohol. Statistically we saw no effect of drinks per pregnancy or number of cigarettes per week on 5-HT1A receptor binding. Although this lack of effect may suggest that alcohol and smoking have limited effects on binding, it may also reflect an inability to meaningfully parse out any relationship between exposure and binding, due to the near-ubiquitous use of cigarettes and alcohol “

• Reviewer #3. “While the study is well-executed and rigorously presented, few clarifications, analyses, and interpretive expansions are needed before publication.”

-We have attempted to identify and clarify the study statements where noted by the reviewers.

• Reviewer #4. “(The manuscript) requires stronger justification and more transparent reporting, especially in relation to multiple comparisons and the modeling of small subgroups.”

-We have added the following statement at the end of the Statistical Methods section. “In all analyses, a p-value≤0.05 was accepted for statistical significance. No formal adjustment for multiple testing was performed, but consistency of results across multiple related outcomes is emphasized.”

-We acknowledge that are sample sizes are relatively small and have noted this in the limitations. Our overall case numbers in this study, however, are in line with other SIDS autopsy studies published by us and others. The premature subsets of SIDS and controls are larger than any other SIDS autopsy study to date which allowed for the analyses of prematurity��analyses considered to be exploratory in nature and warranting replication in larger datasets.

• “The interpretation of exposure-related findings should be more cautious, as the cohort’s near-universal exposure to smoking and alcohol limits the ability to draw meaningful conclusions.

-We agree that we cannot draw meaningful conclusions about the effect of exposure. We have emphasized this point in the limitations section (see comments to Reviewer #1).

• “Additionally, some claims made in the discussion overstate the generalizability of the findings and should be revised for a more measured tone.”

-We have taken a more measured tone on certain statements in the discussion. We have also included the following statement: “While the results of the study are readily generalizable to high-risk population with prenatal smoking and alcohol intake and prematurity, the results suggest that pre- and/or postnatal stress is the common denominator and reflects the burden and type of stress in each population, with the overall nature and burden of the stress being unique in each population. “

Other Peer Review comments: #1.

• Regarding the mention of maternal anxiety and depression in the abstract, “Please, kindly consider specifying p-values or clarifying the statistical significance of these findings”.

-In order to make suggested additions to the abstract while maintaining the necessary word count, we took out the sentence regarding anxiety and depression. We added p-values for other SES proxies.

• “[The introduction] introduces “serotonopathies” appropriately but could briefly clarify the mechanistic importance for non-specialist readers”

-We have added details of the mechanistic importance of serotonin in the 2nd paragraph of the introduction as suggested.

• The text mentions “triple-risk model” and intrinsic vulnerability but doesn’t explain how this informs the sampling or statistical approach. Briefly aligning the cohort selection with this model.

-The Triple risk model is a model or hypothesis of the pathogenesis of SIDS. It does not inform the sampling or statistical approach. This was an autopsy study and all available cases and controls were included in the study if technically usable. Among the cohort available to us for analysis, the three components of the model reflected characteristics of the unique study population and drove the interpretation of our results.

• The rationale for grouping SIDS deaths based on postconceptional age is justified but consider explaining how gestational age was estimated (e.g., clinical records, last menstrual period).

-Gestational age at enrollment was determined during the first prenatal visit using standard clinical practices at each study center – ultrasound in South Africa, and a combination of clinical examination, ultrasound, and last menstrual period in the Northern Plains. This was previously reported in Dukes et al Alcohol. 2017 August; 62: 49–60. This has been added to the text.

• It's unclear if any matching strategy (for postconceptional age) was employed when selecting controls.

-There was no matching of cases with controls. This was an autopsy study, and all available cases were included.

• Although it's noted that cases were excluded if the pons were damaged, it would help to provide the number or percentage of cases excluded for transparency and potential selection bias assessment.

-We have added the following information into the Clinical Database part of the manuscript. “Of the 14 SIDS and 10 PostKCOD controls available and of suitable quality for autoradiography, only 9 (64%) SIDS had rostral pons available but with varying availability of regions of interest within the pons (n=6-9). Four PostKCOD controls (40%) had rostral pons available.

• There is no mention of whether assay batch effects were controlled for (e.g., all samples run in a single batch or randomized across batches).

We added the following statement to the Methods section: “Autoradiography binding was performed in batches with cases blinded to diagnosis and randomly distributed across the different batches. The same autoradiography standard was used across the different batches.”

• Missing data on exposures were imputed as “not exposed” or “not diagnosed.” This assumption is highly problematic and may introduce systematic misclassification bias. If such an assumption was necessary, a sensitivity analysis excluding imputed cases should be performed and results compared

-The method for the imputation is published in multiple studies, cited below. These manuscripts have been added to the text.

Sania A, Pini N, Nelson ME, Myers MM, Shuffrey LC, Lucchini M, Elliott AJ, Odendaal HJ, Fifer WP. K-nearest neighbor algorithm for imputing missing longitudinal prenatal alcohol data. Adv Drug Alcohol Res. 2025 Jan 28;4:13449. doi: 10.3389/adar.2024.13449. PMID: 39935524; PMCID: PMC11811783..

Shuffrey LC, Myers MM, Isler JR, et al; PASS Network. Association between prenatal exposure to alcohol and tobacco and neonatal brain activity: results from the Safe Passage Study. JAMA Netw Open. 2020;3(5):e204714. doi:10.1001/jamanetworkopen.2020.4714

• Testing interactions with PCA despite small sample sizes should be further justified. The authors acknowledge this, but an explanation of why they proceeded (such as effect sizes in previous studies) would strengthen transparency.

-Our analyses were planned a priori and interactions were tested based on previous published work that noted that the effect of age could differ by groups (See references).

Duncan JR, Paterson DS, Hoffman JM, Mokler DJ, Borenstein NS, Belliveau RA, et al. Brainstem serotonergic deficiency in sudden infant death syndrome. JAMA. 2010;303:430-7.

Haynes RL, Trachtenberg F, Darnall R, Haas EA, Goldstein RD, Mena OJ, et al. Altered 5-HT2A/C receptor binding in the medulla oblongata in the sudden infant death syndrome (SIDS): Part I. Tissue-based evidence for serotonin receptor signaling abnormalities in cardiorespiratory- and arousal-related circuits. J Neuropathol Exp Neurol. 2023;82(6):467-82

• Was any multiple comparisons correction (e.g., Bonferroni or FDR) considered given the number of nuclei analyzed?

-Due to the small number of cases, correction for multiple comparisons was not done. Although sample sizes were not amenable to warrant multiple testing corrections, our findings are strengthened by the consistency of results across several nuclei.

• The use of multiple linear regressions for each nucleus increases Type I error risk. Was any correction for multiple testing applied (Bonferroni or FDR)? If not, the risk of false positives should be explicitly discussed in the limitations.

-See the response above.

• k-Nearest Neighbor (kNN) imputation for missing binding values is acceptable, but the number of neighbors (k), the distance metric, and whether cross-validation was used to tune k are not described.

• While it's good to see kNN imputation used, more detail is needed: What was the rationale for choosing kNN over other methods (such as multiple imputation)?

• It's unclear which variables were used in the imputation model, and whether missingness was assumed to be Missing at Random (MAR), Missing Completely at Random (MCAR), or otherwise and whether any sensitivity analysis was done. A brief discussion of imputation assumptions would be useful.

-Please see the response above regarding kNN imputation. This method has been published.

• Given high exposure prevalence to alcohol and smoking in both SIDS and control groups, the authors correctly note that comparisons are underpowered. However, this severely limits conclusions related to prenatal exposures. Recommend emphasizing this limitation earlier in the abstract and results

-We had emphasized this in the original abstract and did not make changes. We have now emphasized this limitation in the Results and more fully describe it in the Limitations section of the Discussion.

• Interaction terms are tested but need clearer interpretation. Where significant diagnosis x age interactions are found, authors should clarify whether slopes differ by group or whether there is a crossover effect.

Our data with significant interaction effects suggests that SIDS is associated with higher binding compared to PostKCOD controls, but only at younger ages and only in preemies. We have added this point to the results section.

• Table 8 shows no significant effect of smoking or alcohol exposure on binding. This contradicts earlier hypothesis and highlights cohort limitations (ubiquitous exposure). Authors acknowledge this, but it should be reiterated in the discussion as a major constraint.

-We have highlighted this in the beginning of the Discussion and in the Limitations section, the latter with additional text added to emphasize this important point.

• Discussed epigenetic mechanisms are speculative and should be clearly stated as hypotheses without overstating evidence

-We have modified the text concerning epigenetics to clarify that it is a hypothesis.

• Claim that “serotonergic circuits are altered in all SIDS cohorts” may be too broad. Suggest qualifying this to reflect the study’s limitations and absence of a truly “healthy” control group.

-We have modified text from the discussion to now read: “While the SPS cohort has limitations (i.e, small numbers, lack of healthy controls, etc), as does any autopsy study, an important point is that medullary serotonergic circuits are altered in all SIDS cohorts that we have studied to date, irrespective of global background.”

Other Peer Review comments: #2.

• Include in the abstract a sentence highlighting the novel finding of increased 5-HT1A binding in premature SIDS infants.

-We have added this to the abstract

• Clarify early that the study was unable to test the influence of maternal smoking/drinking due to high prevalence in both groups.

-This limitation is noted in the abstract. We have attempted to emphasize it more in the Results and Discussion.

• Consider emphasizing earlier into the introduction, that prior studies observed reduced 5-HT1A binding in SIDS to better contrast the present findings.

-This point was previously made in the second paragraph of the Introduction with references to the manuscripts showing this reduction. To maintain the flow of information within the Introduction, we did not move or alter this section.

• Add a brief explanation (maybe a paragraph) of 5-HT1A receptor function in early brain development to frame the broader physiological implications.

-We have added a short summary of the role in 5-HT1A to the section “Potential Consequences in Infant Physiology of a Medullary Serotonopathy in the SPS” in the Discussion.

• Provide a power analysis or justification for the sample size given the number of comparisons.

-Power analyses were done prior to the initiation of the PASS study in 2007. These are described in Dukes et al. The Safe Passage Study: Design, Methods, Recruitment, and Follow-Up Approach. Paediatr Perinat Epidemiol. 2014 September; 28(5): 455–465.

•

---

## [Decision Letter · Decision Letter 1]

8 Aug 2025

Serotonergic Receptor Binding in the Brainstem in the Sudden Infant Death Syndrome in a High-Risk Population

PONE-D-25-13478R1

Dear Dr. Haynes,

We’re pleased to inform you that your manuscript has been judged scientifically suitable for publication and will be formally accepted for publication once it meets all outstanding technical requirements.

Kind regards,

Yusuf Oloruntoyin Ayipo, Ph.D

Academic Editor

PLOS ONE

Additional Editor Comments (optional):

The submission is scientifically sound for publication in this title, and all the concerns raised by the respective reviewers regarding the manuscript quality have been satisfactorily addressed. I hereby recommend the manuscript for publication in the current version.

Reviewers' comments:

Reviewer's Responses to Questions

**Comments to the Author**

1. If the authors have adequately addressed your comments raised in a previous round of review and you feel that this manuscript is now acceptable for publication, you may indicate that here to bypass the “Comments to the Author” section, enter your conflict of interest statement in the “Confidential to Editor” section, and submit your "Accept" recommendation.

Reviewer #1: All comments have been addressed

Reviewer #5: All comments have been addressed

Reviewer #6: All comments have been addressed

2. Is the manuscript technically sound, and do the data support the conclusions?

Reviewer #1: Yes

Reviewer #5: Yes

Reviewer #6: Yes

3. Has the statistical analysis been performed appropriately and rigorously? 

Reviewer #1: Yes

Reviewer #5: Yes

Reviewer #6: Yes

4. Have the authors made all data underlying the findings in their manuscript fully available?

Reviewer #1: Yes

Reviewer #5: Yes

Reviewer #6: Yes

5. Is the manuscript presented in an intelligible fashion and written in standard English?

Reviewer #1: Yes

Reviewer #5: Yes

Reviewer #6: Yes

6. Review Comments to the Author

Reviewer #1: I would like to commend the authors for their careful and thorough revisions, which have clearly addressed the feedback from the previous review round. The manuscript now meets the publication criteria for PLOS ONE.

Strengths of the work include:

1. Rigorous methodology with blinded autoradiography and well-described sample processing.

2. Unique investigation of SIDS in socioeconomically disadvantaged and high-risk populations.

3. Clear acknowledgment of limitations and confounding factors.

4. Important contribution to understanding serotonergic mechanisms underlying SIDS risk, particularly in preterm infants.

Furthermore, I suggest that the authors may consider briefly reiterating the translational significance of elevated 5-HT1A binding in preterm SIDS cases for non-specialist readers.

Reviewer #5: Dear authors,

Thank you for considering the reviewers recomendations and suggestions in your manuscript.

Kind regards

Reviewer #6: (No Response)

7. PLOS authors have the option to publish the peer review history of their article (what does this mean? ). If published, this will include your full peer review and any attached files.

**Do you want your identity to be public for this peer review?** For information about this choice, including consent withdrawal, please see our Privacy Policy .

Reviewer #1: No

Reviewer #5: **Yes: ** Prof. Dr. Yaareb J. Mousa

Reviewer #6: No

---

## [Editor Report · Acceptance letter]

PONE-D-25-13478R1

PLOS ONE

Dear Dr. Haynes,

I'm pleased to inform you that your manuscript has been deemed suitable for publication in PLOS ONE. Congratulations! Your manuscript is now being handed over to our production team.

Kind regards,

on behalf of

Dr. Yusuf Oloruntoyin Ayipo

Academic Editor

PLOS ONE